



# An uncertainty-based protocol for the setup and measurement of soot and black carbon emissions from gas flares using sky-LOSA

**Bradley M. Conrad and Matthew R. Johnson**

Energy and Emissions Research Laboratory, Department of Mechanical and Aerospace Engineering,
Carleton University, Ottawa, ON K1S 5B6, Canada

**Correspondence:** Matthew R. Johnson (matthew.johnson@carleton.ca)

**Abstract.** [TS1]Gas flaring is an important source of atmospheric soot–black[CE1] carbon, especially in sensitive Arctic regions. However, emissions have traditionally been challenging to measure and remain poorly characterized, confounding international reporting requirements and adding uncertainty to climate models. The sky-LOSA optical measurement technique has emerged as a powerful means to quantify flare black carbon emissions in the field, but broader adoption has been hampered by the complexity of its deployment, where decisions during setup in the field can have profound, non-linear impacts on achievable measurement uncertainties. To address this challenge, this paper presents a prescriptive measurement protocol and associated open-source software tool that simplify acquisition of sky-LOSA data in the field. Leveraging a comprehensive Monte Carlo-based general uncertainty analysis (GUA) to predict measurement uncertainties over the entire breadth of possible measurement conditions, general heuristics are identified to guide a sky-LOSA user toward optimal data collection. These are further extended in the open-source software utility, SetupSky-LOSA, which interprets the GUA results to provide detailed guidance for any specific combination of location, date–time, and flare, plume, and ambient conditions. Finally, a case study of a sky-LOSA measurement at an oil and gas facility in Mexico is used to demonstrate the utility of the software tool, where potentially small regions of optimal instrument setup are easily and quickly identified. It is hoped that this work will help increase the accessibility of the sky-LOSA technique and ultimately the availability of field measurement data for flare black carbon emissions.

## 1 Introduction

Gas flaring is a routine practice in the oil and gas industry in which producers and refiners burn excess or unwanted gases in open-atmosphere flames, typically from vertical pipe stacks. Flaring is generally preferable to the venting of gases to the atmosphere because it reduces carbon dioxide ($CO_2$)-equivalent emissions; however, flaring still emits potent climate-forcing pollutants directly to the atmosphere (Allen and Torres, 2011; Johnson et al., 2001, 2011, 2013; McDaniel, 1983; Pohl et al., 1986). These pollutants have public health implications (e.g., Anenberg et al., 2012) and include unburnt hydrocarbons, volatile organic compounds, and particulate matter (U.S. EPA, 2018). Soot particulate matter (commonly referred to as black carbon, BC) has been suggested by some to be the second most potent climate forcer after $CO_2$ (Bond et al., 2013; Jacobson, 2001; Ramanathan and Carmichael, 2008; Sato et al., 2003). Annual flaring is estimated by satellite imagery to be $\sim 140$ billion $m^3$ (Elvidge et al., 2007, 2009, 2016), making it one important source of global soot emissions. Although other industrial sectors dominate gas flaring in absolute soot emissions, the locations of flaring activities (particularly in Russia) likely have a disproportionate impact on the sensitive Arctic climate due to efficient transport pathways penetrating the Arctic air mass (e.g., Popovicheva et al., 2017; Stohl et al., 2007).

With the addition of BC to the Gothenburg protocol in 2012 (United Nations Economic Commission for Europe (UNECE), 2012), 34 countries are now legally bound to report, where data are available, soot–BC emissions under UN-ECE's Convention on Long-Range Transboundary Air Pollution, including the European Union, Russia, the United States

of America, and Canada. To attribute – and, hence, report and regulate – soot–BC emissions from various sources, emission factors that relate soot–BC emissions to a measure of industrial activity are required. Unfortunately, for gas flaring, commonly employed soot emission factors are crude single-valued parameters that link emitted soot mass to volume or mass of gas flared regardless of flare design, gas composition, or operating conditions. This contrasts with numerous studies that have observed a significant influence of flare gas composition and flame aerodynamics on soot emissions (Becker and Liang, 1982; Conrad and Johnson, 2017; McEwen and Johnson, 2012) and even soot properties (Conrad and Johnson, 2019; Trivanovic et al., 2020). Further soot yield data are needed, particularly for real-world flares under field conditions, to develop and validate accurate flare soot–BC emission factor models.

At present, there are only two published methods for the quantitative measurement of soot–BC emissions from individual in-field flares. One technique employs aircraft-based sampling of a flare plume (Gvakharia et al., 2017; Weyant et al., 2016), where measurements of soot, methane, and $CO_2$ concentrations during transects through the plume are used to provide flare-specific estimates of soot yield, using assumed flare gas compositions. The second technique is a ground-based remote optical measurement called sky-LOSA (line-of-sight attenuation using skylight; Conrad and Johnson, 2017; Johnson et al., 2010, 2011, 2013). Sky-LOSA quantifies time-resolved soot mass emission rates through analysis of high-speed image data. Parallel access to flare infrastructure permits simultaneous measurement of flare gas flow rate and gas sample extraction for off-site compositional analysis, which enables the *direct* calculation of soot yield for a targeted flare. To date, sky-LOSA has been deployed on 11 field measurement campaigns in Uzbekistan, Mexico, Ecuador, and Canada, providing 28 measurements of soot emissions from 17 unique flares (Conrad and Johnson, 2017; Johnson et al., 2011, 2013).

The key component of a sky-LOSA measurement is the quantification of plume soot loading using image data, via analysis and modelling of radiative transfer through the atmospheric flare plume at the measurement wavelength. For each acquired image, soot mass column density is resolved pixel by pixel over a control surface within the image plane to permit mass emission rate calculation CE2. Uncertainties in sky-LOSA-calculated emission rate are computed under a Monte Carlo (MC) framework and are dominated by uncertainties that affect computation of soot mass column density. While these uncertainties are influenced by numerous parameters considered within the MC analysis, they are also sensitive to the positioning and pointing of the sky-LOSA camera relative to the horizon and sun. Consequently, a sky-LOSA user must position the camera according to several constraints, which may be heuristic but can also vary with uncontrollable measurement parameters. To make the measurement technique accessible to end-users, enabling an increase

in flare soot emissions data, a standardized data acquisition protocol for sky-LOSA is required.

The objective of this work is to complete a general uncertainty analysis (GUA) for the sky-LOSA measurement technique that provides uncertainty-based guidance to an end-user regarding the setup of equipment and acquisition of sky-LOSA data through an accompanying open-source software tool. A summary of sky-LOSA theory, referring to derivations in previous works, is first provided in Sect. 2 of this paper. The GUA methodology is summarized in Sect. 3, including special provisions necessary to reduce the computational burden of the MC-based approach (Sect. 3.1 and Appendix A). Representative results from the MC GUA are shown in Sect. 4.1, and general heuristics for the acquisition of sky-LOSA data, including new observations based on MC GUA results, are summarized in Sect. 4.1.1 and 4.1.2. To provide case-by-case guidance, a new open-source software tool to calculate sky-LOSA measurement uncertainty is introduced in Sect. 4.2. Finally, in Sect. 4.3, the software tool is used in a case study that analyzes optimal camera positioning for flare measurements at a gas refining and transport facility in Campeche, Mexico. This work enables a consistent approach for the selection of sky-LOSA camera positioning and pointing to minimize measurement uncertainties, ultimately contributing to the standardization of the sky-LOSA measurement technique.

## 2  Sky-LOSA measurement

The generalized sky-LOSA theory was summarized in full by Johnson et al. (2013) and has been the subject of a variety of validation efforts (Conrad et al., 2020a, b; Johnson et al., 2010). Development of the theory begins with Fig. 1, which shows an example sky-LOSA image for computation of time-resolved soot emission rate from a soot-laden flare plume in the Montney formation of Alberta, Canada. A highly linear, grayscale, scientific-CMOS camera (e.g., pco edge 5.5) is used to obtain upwards of 10 min of high-speed image data of the flare and turbulent, soot-laden, atmospheric plume. Pseudo 16-bit images are acquired at frame rates of 25–50 Hz with a narrow mid-visible bandpass filter ($531 \pm 20$ nm) to yield a scene of spectrally integrated light intensity.

Overlaid in Fig. 1 is an example control surface ($C$) of specified radius ($r$ [m]), through which the instantaneous mass emission rate of soot ($\dot{m}_s$ [g s$^{-1}$]) may be computed. For an arbitrary control surface in three dimensions, the instantaneous mass flux of soot through the surface is

$$\dot{m}_s = \iint\limits_A (\rho_s \boldsymbol{u}) \cdot \boldsymbol{n} \mathrm{d}A, \tag{1}$$

where $\rho_s$ is the mass concentration of soot [g m$^{-3}$], **u** is the local plume velocity vector [m s$^{-1}$], **n** is the unit vector locally normal to the control surface [–], and d$A$ is an infinitesimal area [m$^2$]. For sky-LOSA, where three-dimensional data

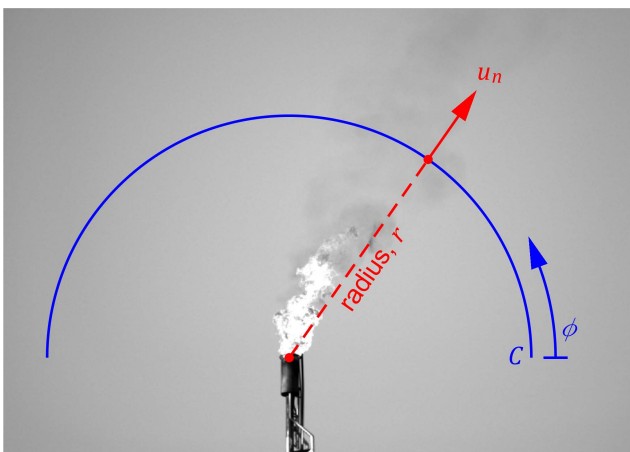

**Figure 1.** Sample sky-LOSA image of the flare and atmospheric plume, which is under slight crosswind in this example. A control surface ($C$) is shown in blue, which is defined by its constant radius ($r$) and the angle $\phi$. At each point on the control surface, the mass column density ($\rho_s'(r, \phi)$, not shown) is computed via careful consideration of radiative effects along the lines of sight (perpendicular to the image plane) that compose the control surface. Additionally, the path-averaged normal plume velocity ($u_n(r, \phi)$) is computed via image correlation velocimetry. The instantaneous mass emission rate is computed by integrating the product of these over the control surface defined by $r\mathrm{d}\phi$.

along a pixel's line of sight (LOS) are collapsed to two dimensions through projection, an equivalent formulation for the instantaneous mass emission rate is

$$\dot{m}_s = \int_C \rho_s'(r, \phi) \, u_n(r, \phi) \, r \, \mathrm{d}\phi, \qquad (2)$$

where $\rho_s'(r, \phi) = \int \rho_s(r, \phi, x) \, \mathrm{d}x$ is the soot mass column density along a LOS (where the $x$ dimension is orthogonal to the image plane) [g m$^{-2}$] and $u_n(r, \phi) = \left( \int \rho_s(r, \phi, x) \, \mathbf{u}(r, \phi, x) \cdot \mathbf{n}(\phi) \, \mathrm{d}x \, / \, \rho_s'(r, \phi) \right)$ represents the component of the mass-concentration-weighted
LOS-averaged velocity of the plume [m s$^{-1}$] that is normal to the control surface from the camera's perspective (as shown in the figure). Via Eq. (2), sky-LOSA thus requires knowledge of three items to compute the emission rate: spatial scaling of the image plane to accurately quantify
$r$, the velocity field of the plume within the image plane (yielding $u_n(r, \phi)$), and the soot mass column density resolved over the control surface. Spatial scaling of the image is obtained through use of a pinhole analogy for the sky-LOSA optics, coupled with a measurement of the
distance to the flare stack tip by laser rangefinder (e.g., Laser Technology Inc. TruPulse 360R; Johnson et al., 2013). Given that imaging is performed with a global shutter and at a sufficiently rapid frame rate and exposure, the two-dimensional plume velocity field over the image plane
is estimated via image correlation velocimetry, using a third-party software suite such as LaVision DaVis 8.4 that

includes a means of uncertainty quantification (Wieneke, 2015). Finally, the novel enabling aspect of sky-LOSA is the use of bounded knowledge of soot optical properties from literature data to compute the soot mass column density 30 with accurate uncertainties via radiometric observations and modelling of radiative transfer within the atmosphere and plume. Example sky-LOSA measurements of time-resolved soot emission rate are available in previous works (Conrad and Johnson, 2017; Johnson et al., 2011, 2013). 35

Figure 2 shows an example positioning and pointing of the sky-LOSA camera and an optical axis and (LOS) within the surrounding sky dome. For a given LOS, a Cartesian coordinate system is defined where the positive $x$ direction is the path that light travels into the camera, the positive $z$ direction 40 is the general direction of plume motion, and the $y$ dimension completes the orthogonal system. To model radiative transfer, there are three boundary conditions that must be considered. Firstly, the ground is treated as a cold, black surface and is thus ignored within the sky-LOSA algorithm. Secondly, 45 the sky is modelled as a diffuse, polarized source concomitant with atmospheric scattering of solar radiation. The distribution of skylight intensity ($I(\alpha, \alpha_s, Z, Z_s, a)$ [W m$^{-2}$ sr$^{-1}$]) and the incident intensity along the LOS ($I^o(\alpha_s, \beta, Z_s, a)$ [W m$^{-2}$ sr$^{-1}$]) are considered using the standard models of 50 the Commission Internationale de l'Eclairage (CIE, 2003), where the index $a \in \{1 \ldots 15\}$ indicates a specific CIE sky "type". Finally, ground-level normal solar irradiance ($E_{sn}$ [W m$^{-2}$]) is estimated using in-field, image-based measurements of the sun (Johnson et al., 2013) or modelled using the 55 CIE models. With this radiative transfer model, sky-LOSA quantification of soot mass column density proceeds with the radiative transfer equation (RTE):

$$I^t = I^o \exp\left( -\sigma_m^e(\boldsymbol{b}) \int_{-\infty}^{L} \rho_s(x) \, \mathrm{d}x \right)$$
$$+ \int_{-\infty}^{L} J^s(x, \boldsymbol{b}) \, \rho_s(x) \exp\left( -\sigma_m^e(\boldsymbol{b}) \int_{x}^{L} \rho_s(x') \, \mathrm{d}x' \right) \mathrm{d}x, \qquad (3)$$

where $I^t$ is the measured "transmitted" intensity at the camera [W m$^{-2}$ sr$^{-1}$], $\sigma_m^e(\boldsymbol{b})$ is the mass-normalized extinction 60 cross section of the polydisperse soot population [m$^2$ g$^{-1}$] that is a function of eight soot properties represented by the vector $\boldsymbol{b}$, $J^s(x, \boldsymbol{b})$ is a local source radiant intensity per unit mass [W sr$^{-1}$ g$^{-1}$] along the measurement path, and the sky- 65 LOSA camera is located at $x = L$. Population-averaged optical properties of soot are computed from the fundamental properties in $\boldsymbol{b}$ using Rayleigh–Debye–Gans theory for polydisperse fractal aggregates (RDG-PFA; e.g., Sorensen, 2001). The vector $\boldsymbol{b}$ is composed of the absorption function 70 of soot at the measurement wavelength ($E(m_\lambda)$ [–], where $m_\lambda$ is the spectral complex refractive index of soot [–]), the scattering-to-absorption function of soot at the measurement wavelength ($F(m_\lambda)/E(m_\lambda)$ [–]), the monodisperse primary

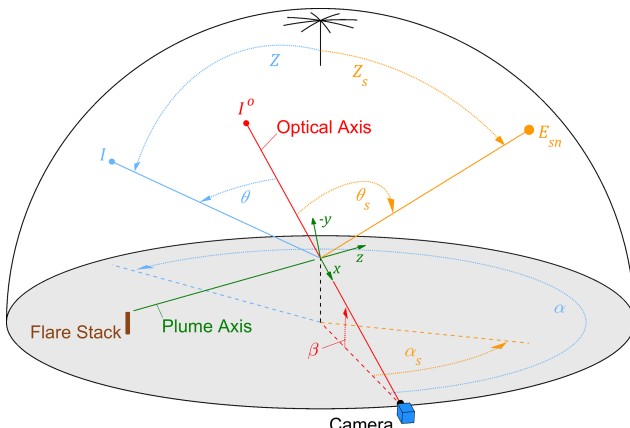

**Figure 2.** Schematic of a sky-LOSA measurement under the hemispherical sky dome showing the camera's optical axis relative to the horizon ($\beta$), sun ($\alpha_s$, $\theta_s$, and $Z_s$), and example sky element ($\alpha$, $\theta$, $Z$) (adapted from Conrad et al., 2020b).

particle diameter of the soot population ($d_p$ [nm]), the geometric mean soot aggregate size ($N_g$ [–]), the geometric standard deviation of the lognormal soot aggregate size distribution ($\sigma_g$ [–]), the *diameter-based* fractal prefactor ($k_f$ [–]), the fractal (Hausdorff) dimension ($D_f$ [–]), and the material density of soot ($\varrho_s$ [g cm$^{-3}$]). Consistent with laboratory observations of soot structure in the overfire region of flare-like flames (e.g., Köylü and Faeth, 1992), these eight properties are assumed to be spatially and temporally uniform within a *single* MC draw; however, they are treated as random variables within the sky-LOSA MC method (Johnson et al., 2013). The prior probability distributions employed within the MC method inherently link light absorption measurements and computed mass column density and emissions using sky-LOSA; as such, in keeping with Andreae and Gelencsér (2006) and Petzold et al. (2013), sky-LOSA-inferred soot–BC mass might therefore be called "equivalent BC" as is recommended for all light-absorption-based diagnostics, especially where absorption-enhancing non-BC material may be present. For the case of flare-generated soot–BC however, studies in the field (Schwarz et al., 2015; Weyant et al., 2016) have identified that the presence of non-BC aerosols in flare plumes is "not statistically different from zero" (Weyant et al., 2016), which is supported by laboratory observations (e.g., Kazemimanesh et al., 2019). This justifies use of soot property probability distributions derived from literature data of *freshly emitted* soot particulate by Johnson et al. (2013).

Using the mean value theorem, a path-averaged source radiant intensity ($\overline{J^s}(\boldsymbol{b})$ [W sr$^{-1}$ g$^{-1}$]) can be introduced to

simplify the RTE:

$$
I^t = I^o \exp\left(-\sigma_m^e(\boldsymbol{b}) \underbrace{\int_{-\infty}^{L} \rho_s(x)\mathrm{d}x}_{\tau^*}\right)
$$

$$
+ \underbrace{\frac{\overline{J^s}(\boldsymbol{b})}{\sigma_m^e(\boldsymbol{b})}}_{\overline{I^s}} \underbrace{\int_{\infty}^{L} \sigma_m^e(\boldsymbol{b})\rho_s(x)\exp\left(-\sigma_m^e(\boldsymbol{b})\int_{x}^{L}\rho_s(x')\mathrm{d}x'\right)\mathrm{d}x}_{1-\tau^*,} \quad (4)
$$

where $\sigma_m^e(\boldsymbol{b})/\sigma_m^e(\boldsymbol{b}) = 1$ has also been introduced to the source term. The resulting ratio in front of the second integral represents a path-averaged source intensity ($\overline{I^s}$ [W m$^{-2}$ sr$^{-1}$]), and the exponential in the first term corresponds to the transmittance of the plume in the absence of radiative sources, defined as the idealized transmittance $\tau^*$ (Johnson et al., 2013). It can be shown that the integral in the second term is equal to the complement of the idealized transmittance, permitting solution of the latter from the three intensities in Eq. (4):

$$
\tau^* = \frac{I^t - \overline{I^s}}{I^o - \overline{I^s}} = \frac{\tau_{obs} - S}{1 - S}, \quad (5)
$$

where $\tau_{obs} = I^t/I^o$ is the observed transmittance of the plume [–] and $S = \overline{I^s}/I^o$ is a term that corrects for brightening of the plume by radiative sources [–]. Noting the definition of the idealized transmittance, the column density of soot is simply

$$
\rho_s'(\tau_{obs}, S, \boldsymbol{b}) = \frac{-\ln\left(\frac{\tau_{obs}-S}{1-S}\right)}{\sigma_m^e(\boldsymbol{b})}. \quad (6)
$$

In general, the local source radiant intensity ($J^s(x, \boldsymbol{b})$) is composed of thermal emission and inscattering components. For sky-LOSA however, where measurements are performed in the mid-visible spectrum and plume temperatures are near ambient, thermal emission negligibly contributes to the source term of the RTE (Conrad et al., 2020b). By contrast, diffuse skylight and direct solar radiation can significantly augment the RTE via inscattering into the optical axis; hence, for sky-LOSA, $S$ represents an *inscattering correction*. Unfortunately, even if exact knowledge of skylight intensity distribution and solar radiation were available, it is not possible to fully account for the effect of inscattering without prior knowledge of the spatial distribution of soot within the plume. This is because multiple-scattering events may occur during light's transmission into the camera. A recent simulation effort, however, has shown that complex multiple-scattering effects can be accurately modelled to permit quantification of the inscattering correction ($S$) (Conrad et al., 2020b). This approach requires calculation of the inscattering correction using a *single-scattering assumption* (1SA,

subscript "1"):

$$S_1(\alpha_s, \beta, Z_s, a, \boldsymbol{b}) =$$

$$\underbrace{\int_0^{2\pi} \int_0^{\pi/2} \frac{I(\alpha, \alpha_s, Z, Z_s, a)}{I^{\circ}(\alpha_s, \beta, Z_s, a)} \frac{\omega(\boldsymbol{b})}{4\pi} p(\theta(\alpha, \beta, Z), \boldsymbol{b}) \sin Z \, dZ \, d\alpha}_{S_{1,\text{sky}}(\alpha_s, \beta, Z_s, a, \boldsymbol{b})}$$

$$+ \underbrace{\frac{E_{\text{sn}}(Z_s, a)}{I^{\circ}(\alpha_s, \beta, Z_s, a)} \frac{\omega(\boldsymbol{b})}{4\pi} p(\theta_s(\alpha_s, \beta, Z_s), \boldsymbol{b})}_{S_{1,\text{sun}}(\alpha_s, \beta, Z_s, a, \boldsymbol{b}),} \qquad (7)$$

where $\omega(\boldsymbol{b})$ is the single-scattering albedo of the polydisperse soot population [–], $p(\theta; \boldsymbol{b})$ is the scattering phase function of soot [$\text{sr}^{-1}$], and angles $\theta$ and $\theta_s$ represent the angles between the LOS and a region of sky or the sun as shown in Fig. 2. The inscattering correction can be parsed into sky ($S_{1,\text{sky}}$) and sun ($S_{1,\text{sun}}$) components. Following Conrad et al. (2020b), the 1SA-estimated inscattering correction permits calculation of the idealized transmittance – i.e., $\tau^* = \tau^*(\tau_{\text{obs}}, S_1(\alpha_s, \beta, Z_s, a, \boldsymbol{b}))$. Ultimately, this allows for calculation of the soot mass column density by

$$\rho_s'(\alpha_s, \beta, Z_s, \tau_{\text{obs}}, a, \boldsymbol{b}) = \frac{-\ln \tau^*(\tau_{\text{obs}}, S_1(\alpha_s, \beta, Z_s, a, \boldsymbol{b}))}{\sigma_{\text{m}}^{\text{e}}(\boldsymbol{b})}. \qquad (8)$$

According to Eq. (8), sky-LOSA computation of soot mass column density is a function of the position of the camera and sun, field-observed plume transmittance, skylight intensity distribution and solar irradiance (through $a$), and soot properties. In the sky-LOSA algorithm, soot mass column density is directly calculated by MC analysis over uncertain variables, which include soot properties and other intermediate parameters. This implies that *uncertainty* in sky-LOSA-computed soot mass column density is sensitive to $\alpha_s$, $\beta$, $Z_s$, $\tau_{\text{obs}}$, and $a$. Of these five variables, only the pointing of the camera may be controlled by the user, through the selection of angles $\alpha_s$ and $\beta$. Generally, overall uncertainty in sky-LOSA-computed soot mass column density is driven by uncertain soot properties through their effect on the mass-normalized extinction cross section and the single-scattering estimate of the inscattering correction. Under optimal camera positioning, the latter is negligible relative to the former, and 95 % confidence intervals on sky-LOSA-computed soot mass emission rates are on the order of $-26\,\%/+36\,\%$ (Conrad and Johnson, 2017; Johnson et al., 2013). However, the magnitude and uncertainty of the inscattering correction are strongly sensitive to the user-selected angles, $\alpha_s$ and $\beta$, and, in extreme cases, can even preclude accurate computation of the soot emission rate.

For end-users of sky-LOSA, the sensitivity of measurement uncertainty to camera pointing necessitates a standardized (and ideally simple) data acquisition protocol to optimize camera position and pointing under general conditions.

This would allow a priori setup decisions to minimize or constrain uncertainties within reasonable limits. An acquisition protocol must therefore be constructed using quantitative knowledge of measurement uncertainty in sky-LOSA-computed soot mass column density. Restated in the context of the above theory, the objective of this work is to quantify via a comprehensive general uncertainty analysis the uncertainty in sky-LOSA computation of soot mass column density ($\rho_s'$) as a function of user-selectable ($\alpha_s$ and $\beta$) and uncontrollable parameters ($Z_s$, $\tau_{\text{obs}}$, and $a$) under generalized conditions. These data permit the development of broad heuristics and, ultimately, an easy-to-use software tool to provide specific case-by-case constraints on camera position and pointing.

## 3 General uncertainty analysis methodology

The goal of the present general uncertainty analysis (GUA) is to guide a sky-LOSA user in choosing a sky-LOSA camera position and pointing to minimize measurement uncertainties. The developed software tool can also be used to give an initial estimate of uncertainties in the measured soot emission rate ahead of a more detailed post-processing analysis. To provide generalized recommendations, the GUA quantifies measurement uncertainty in soot mass column density for a selected camera pointing and other independent variables via MC analysis over uncertain variables that include all relevant soot properties. This section describes the MC method used in the present GUA including novel updates to the MC approach that are necessary to make this present work tractable. This new methodology is a significant improvement to the sky-LOSA algorithm that enables accelerated MC computation of soot column density and, hence, emission rates from sky-LOSA image data.

### 3.1 Updated sky-LOSA MC method

#### 3.1.1 Expansion of the scattering phase function

For a given (modelled) skylight intensity distribution, measured or modelled solar irradiance, camera pointing, and set of soot properties, the 1SA-estimated inscattering correction ($S_1$) can be directly calculated via Eq. (7). One significant challenge, and currently the time-limiting computation in sky-LOSA processing, is the calculation of the skylight component ($S_{1,\text{sky}}$) via numerical integration over three dimensions: $\alpha$, $Z$, and $N$, where the latter represents the aggregate size distribution of the soot population. The complexity of this task is exacerbated by the scattering phase function (SPF, $p(\theta, \boldsymbol{b})$) of soot, which includes a computationally burdensome hypergeometric series in its solution (Sorensen, 2001). For the present GUA, where the inscattering correction must be computed over a five-dimensional domain ($\alpha_s, \beta, Z_s, a, \boldsymbol{b}$), an alternative, more rapid means of

computing the inscattering correction was required to avoid combinatorial explosion.

One such means is through a Fourier–Legendre expansion of the SPF. For an arbitrary set of randomized soot properties $\boldsymbol{b}$, this procedure allows the SPF to be represented as a weighted sum of Legendre polynomials ($P_l(\cos\theta)$ [–]):

$$p(\theta,\boldsymbol{b}) = \sum_{l=0}^{\infty} \Phi_l(\boldsymbol{b}) P_l(\cos\theta) \approx \sum_{l=0}^{L(\boldsymbol{b})} \Phi_l(\boldsymbol{b}) P_l(\cos\theta), \quad (9)$$

where $L(\boldsymbol{b})$ is the order at which the expansion is truncated [–] and $\Phi_l(\boldsymbol{b})$ is the $l$th-order Legendre coefficient [$\text{sr}^{-1}$] (hereinafter termed the $l$th-order *soot coefficient*) for the set of soot properties ($\boldsymbol{b}$) computed via

$$\Phi_l(\boldsymbol{b}) = \frac{2l+1}{2} \int_0^\pi p(\theta,\boldsymbol{b}) P_l(\cos\theta) \sin\theta \mathrm{d}\theta, \quad (10)$$

which can be accurately and efficiently computed using Gauss–Legendre quadrature (Schuster, 2004). Since the soot coefficient decreases towards zero as the order $l$ approaches infinity, the infinite series expansion of the SPF can be truncated at a sufficiently large index $L(\boldsymbol{b})$ with negligible error (refer to Appendix A Sect. A1 for further details).

Introduction of the Fourier–Legendre-expanded SPF into the sky and sun components of the inscattering correction ($S_{1,\text{sky}}$ and $S_{1,\text{sun}}$, Eq. 7) yields

$$S_{1,\text{sky}}(\alpha_\text{s}, \beta, Z_\text{s}, a, \boldsymbol{b}) =$$
$$\frac{\omega(\boldsymbol{b})}{4\pi} \sum_{l=0}^{L(\boldsymbol{b})} \Phi_l(\boldsymbol{b})$$
$$\underbrace{\int_0^{2\pi} \int_0^{\frac{\pi}{2}} \frac{I(\alpha,\alpha_\text{s},Z,Z_\text{s},a)}{I^\text{o}(\alpha_\text{s},\beta,Z_\text{s},a)} P_l(\cos\theta(\alpha,\beta,Z)) \sin Z \mathrm{d}Z \mathrm{d}\alpha}_{\Psi_{1,\text{sky}}(\alpha_\text{s},\beta,Z_\text{s},a),} \quad (11)$$

$$S_{1,\text{sun}}(\alpha_\text{s}, \beta, Z_\text{s}, a, \boldsymbol{b}) =$$
$$\frac{\omega(\boldsymbol{b})}{4\pi} \sum_{l=0}^{L(\boldsymbol{b})} \Phi_l(\boldsymbol{b})$$
$$\underbrace{\frac{E_\text{sn}(Z_\text{s},a)}{I^\text{o}(\alpha_\text{s},\beta,Z_\text{s},a)} P_l(\cos\theta_\text{s}(\alpha_\text{s},\beta,Z_\text{s}))}_{\Psi_{1,\text{sun}}(\alpha_\text{s},\beta,Z_\text{s},a)}, \quad (12)$$

where $\Psi_{1,\text{sky}}(\alpha_\text{s},\beta,Z_\text{s},a)$ and $\Psi_{1,\text{sun}}(\alpha_\text{s},\beta,Z_\text{s},a)$ are denoted as the *sky* and *sun coefficients*, respectively. Importantly, these equations show that use of the expanded SPF removes reference to soot properties from the integral in the computation of $S_{1,\text{sky}}$, which vastly reduces computational burden. Furthermore, with this formulation, the soot coefficients (functions of $\boldsymbol{b}$) and sky and sun coefficients (functions of $\alpha_\text{s}$, $\beta$, $Z_\text{s}$, and $a$) do not share any independent variables and can therefore be independently pre-computed.

**Table 1.** Sky categories derived to propagate error in the CIE sky models through the sky-LOSA algorithm computing soot mass column density.

| Sky category | CIE sky types ($a$) | Description |
|---|---|---|
| A | 1–6 | Overcast and partly cloudy skies, obscured sun |
| B | 7–10 | Partly cloudy skies, unobscured sun |
| C | 11–15 | Clear skies, all |
| D | 11, 13–15 | Clear skies, polluted |

While the incident intensity-normalized solar horizontal irradiance ($E_\text{sn}(Z_\text{s},a)/I^\text{o}(\alpha_\text{s},\beta,Z_\text{s},a)$ [sr]) is typically measured in the field by obtaining neutral density-filtered images of the sun (Johnson et al., 2013), this parameter must be modelled for the GUA. This was accomplished using the CIE sky models and model-dependent typical turbidity factors and diffuse-to-extraterrestrial solar horizontal irradiance ratios as further detailed in Appendix A Sect. A2.

### 3.1.2 Sky model categorization

The standard CIE sky models have found good utility in a variety of fields, from urban planning (e.g., Acosta et al., 2014) to building design (e.g., Wong, 2017); however, the models naturally suffer from directionally dependent error in skylight intensity. This is particularly true for overcast and partly cloudy skies since the models, which are smooth functions, do not capture steep gradients in skylight intensity due to cloud structures. Thus, there is some additional uncertainty in sky-LOSA-computed soot mass column density through use of a single CIE sky model in the MC method. To permit capture of CIE sky model error in the GUA, like skies were sorted into sky "categories" that have similar properties but differing model coefficients and, hence, directional variability. The derived sky categories ($a \in \{A\ldots D\}$) are summarized in Table 1 and can be selected by a sky-LOSA user based on simple observations in the field such as the visibility of the sun (obstructed vs. unobstructed) and presence of clouds (overcast, partly cloudy, or clear). By randomly selecting a sky category's component sky models under the MC framework, uncertainty through use of the CIE models is propagated into sky-LOSA computation of soot mass column density.

Sky category A corresponds to overcast and partly cloudy conditions with an obscured sun. Typical turbidity factors of the component skies ($T(a)$) are high in these conditions, corresponding to low ground-level solar irradiance. Sky category B represents partly cloudy conditions with an unobscured sun where turbidity factors of the component skies are moderate and hemispherical skylight is strong relative to

extraterrestrial solar radiation. Sky categories C and D capture the low-turbidity clear CIE sky models. Sky category C includes all five clear-sky models, while sky category D excludes CIE sky type 12, the lowest-turbidity (i.e., cleanest atmosphere) model. Sky category D was defined based on the notion that oil and gas activities can be relatively dense geographically. In this case, field experience suggests that local emissions from industrial infrastructure likely preclude CIE sky type 12 as a reasonable model, such that sky category D can be used for the case of clear skies in heavily industrial locales.

### 3.2 MC implementation

Table 2 summarizes the independent, pre-computed, and random variables required to compute soot mass column density under the GUA MC framework. There are five independent variables that define the pointing of the camera relative to the sun ($\alpha_s$, $\beta$, $Z_s$), the observed plume transmittance ($\tau_{obs}$), and the skylight intensity distribution (CIE sky type, $a$). Each MC draw randomly chooses soot properties ($\boldsymbol{b}$) and the scalar multiplier to the sun component of inscattering ($\xi_{ED}(a)$, described in Appendix A Sect. A2). For analyses using the defined sky categories A–D, one CIE sky model is randomly obtained (i.e., $a_k$) from the selected sky category. The $k$th MC estimate of the soot mass column density is then calculated via

$$\rho'_{s,k}(\alpha_s, \beta, Z_s, \tau_{obs}, a_k, \boldsymbol{b}_k) =$$
$$\frac{-\ln \tau^*(\tau_{obs}, S_{1,k}(\alpha_s, \beta, Z_s, a_k, \boldsymbol{b}_k))}{\sigma_m^e(\boldsymbol{b}_k)}, \tag{13}$$

where $\tau^*(\tau_{obs}, S_{1,k}(\alpha_s, \beta, Z_s, a_k, \boldsymbol{b}_k))$ is deterministically computed while considering multiple-scattering effects, as described by Conrad et al. (2020b), and

$$S_{1,k}(\alpha_s, \beta, Z_s, a_k, \boldsymbol{b}_k) = \frac{\omega(\boldsymbol{b}_k)}{4\pi} \sum_{l=0}^{L(\boldsymbol{b}_k)} \Phi_l(\boldsymbol{b}_k)$$
$$(\Psi_{l,sky}(\alpha_s, \beta, Z_s, a_k)$$
$$+ \xi_{ED,k}(a_k) \Psi_{l,sun}(\alpha_s, \beta, Z_s, a_k)). \tag{14}$$

In a standard MC analysis, the above procedure would be iterated upon $K$ times to yield a collection of soot mass column density estimates from which a posterior distribution of soot mass column density could be computed. To accelerate MC procedures in this work, a MC variance reduction technique was employed – specifically, combined multiple Latin hypercube sampling summarized by Nakayama (2011). This variance reduction technique has been used previously for sky-LOSA (Conrad and Johnson, 2017) and was found to reduce computational burden by a factor of 2–3. For the GUA, $5 \times 10^5$ (500 sets of 1000 Latin-hypercube-sampled data) MC draws were completed. The GUA MC approach permitted pre-computation of the soot coefficients, single-scattering albedo, and mass-normalized extinction cross sec-

tion for pre-drawn random sets of soot properties ($\boldsymbol{b}$). Parallel pre-computation of the sky and sun coefficients was performed for each of the 15 CIE standard skylight intensity distributions and four sky categories over the angles $\alpha_s$, $\beta$, and $Z_s$ in increments of 2° and for 18 observed transmittances from 0.25–0.99. This amounted to execution of the MC analysis for almost $66 \times 10^6$ unique sky-LOSA conditions, permitting derivation of uncertainty statistics over the five independent variables listed in Table 2.

To enable an objective comparison of sky-LOSA uncertainty as a function of the independent variables, a parameter describing the *relative uncertainty* of MC-computed soot mass column density was required. A natural means of representing relative uncertainty is a coefficient of variation (CV)-like metric that describes a measure of data variance normalized by a measure of central tendency. For consistency with sky-LOSA measurements, a CV estimator based on the mean and 95 % confidence interval was selected for this work. For variability, the width of the 95 % CI was scaled by that of the standard normal distribution ($\approx 3.92$) to yield an *equivalent* standard deviation, while the mean ($\overline{x}$) was employed as the measure of central tendency. The CV estimator for soot mass column density (or any MC-computed data $\boldsymbol{x}$) was therefore

$$CV_{95}(\boldsymbol{x}) = \frac{\mathcal{F}_x^{-1}(0.975) - \mathcal{F}_x^{-1}(0.025)}{3.92\overline{x}}, \tag{15}$$

where subscript "95" signifies use of the 95 % CI for variability, and $\mathcal{F}_x^{-1}(q)$ is the $q$th quantile of data vector $\boldsymbol{x}$.

## 4 Results and discussion

### 4.1 Representative results

Figure 3 shows relative uncertainty results at different camera pointings for an example sky-LOSA measurement scenario of a flare with 90 % observed plume transmittance. The selected solar zenith angle ($Z_s = 32.8°$) represents the annual minimum for the Canadian city of Fort St. John, British Columbia, which is located in the Montney oil- and gas-producing formation. $CV_{95}$ data are plotted for sky categories A–D in Figs. 3a–d, respectively, as a function of relative solar azimuth ($\alpha_s$ in Fig. 2) and camera inclination ($\beta$); the position of the sun in $\alpha_s$–$\beta$ coordinates (180°, 57.2° TS2) is overlaid in each subfigure. Additionally, contours displaying the solar scattering angle ($\theta_s$) are overlaid in Fig. 3d.

There are two observable trends in the data of Fig. 3 that persist through all measurement conditions. Firstly, the relative uncertainty is a strong function of the solar scattering angle, $\theta_s$. This is because of solar radiation's influence on $S_{1,sky}$ and $S_{1,sun}$. However, the rate at which relative uncertainty changes as a function of $\theta_s$ is dependent on the sky model (as well as solar zenith and plume transmittance). This is partly a consequence of how $S_{1,sky}$ varies with $\theta_s$ but is mostly due to variability in $S_{1,sun}$ with $\theta_s$. For example, the inscattering magnitude for sky category A is effectively due to $S_{1,sky}$

**Table 2.** Summary of independent, random, and pre-computed variables in the GUA.

| Variable group | Variable name | | Symbol | Unit | Source[a] | GUA MC implementation[b] |
|---|---|---|---|---|---|---|
| Ambient lighting | Sky model coefficients | | $a_k$ | [–] | | Independent/ random[c] |
| | Skylight intensity | | $I(\alpha, \alpha_s, Z, Z_s, a)$ | [W m$^{-2}$ sr$^{-1}$] | CIE models | |
| | Incident skylight intensity | | $I^o(\alpha_s, \beta, Z_s, a)$ | [W m$^{-2}$ sr$^{-1}$] | | |
| | Diffuse horizontal irradiance | | $D_h(Z_s, a)$ | [W m$^{-2}$] | Eq. (A2) | |
| | Diffuse-to-extraterrestrial solar horizontal irradiance ratio | | $\frac{D_h(Z_s,a)}{E_{sh,o}(Z_s,a)}$ | [–] | Typical values[d] | Pre-computation of sky and sun coefficients |
| | Turbidity factor | | $T(a)$ | [–] | | |
| | Relative air mass | | $m(Z_s)$ | [–] | Eq. (A4) | |
| | Ideal (clean atmosphere) extinction | | $\sigma^{e*}(m)$ | [–] | | |
| | Sky coefficients | | $\Psi_{l,sky}(\alpha_s, \beta, Z_s, a)$ | [–] | Eq. (11) | |
| | Sun coefficients | | $\Psi_{l,sun}(\alpha_s, \beta, Z_s, a)$ | [–] | Eq. (12) | |
| | Irradiance ratio scaling | | $\xi_{ED}(a)$ | [–] | $\sim \mathcal{U}(0.00, 1.25)$ or $\sim \mathcal{U}(0.75, 1.25)^e$ | Random |
| Soot properties | Soot properties[f] ($\boldsymbol{b}$) | Absorption function of the soot refractive index (at 531 nm) | $E(m_\lambda)$ | [–] | $\sim \mathcal{N}(0.332, 0.0439)$ | |
| | | Scattering-to-absorption function of the soot refractive index (at 531 nm) | $F(m_\lambda)/E(m_\lambda)$ | [–] | $\sim \mathcal{N}(0.901, 0.128)$ | |
| | | Soot monomer/primary particle diameter | $d_p$ | [nm] | $\sim \mathcal{N}(36, 5.81)$ | |
| | | Geometric mean of lognormal aggregate size distribution | $\mathcal{N}_g$ | [–] | $\sim \mathcal{U}(30, 300)$ | Random |
| | | Geometric standard deviation of lognormal aggregate size distribution | $\sigma_g$ | [–] | $\sim \mathcal{N}(2.74, 0.499)$ | |
| | | Fractal prefactor (diameter-based) | $k_f$ | [–] | $\sim \mathcal{N}(8.145, 0.451)$ | |
| | | Fractal (Hausdorff) dimension | $D_f$ | [–] | $\sim \mathcal{N}(1.74, 0.0574)$ | |
| | | Soot material density | $\varrho_s$ | [g cm$^{-3}$] | $\sim \mathcal{N}(1.86, 0.120)$ | |
| | Mass-normalized extinction cross section | | $\sigma_m^e(\boldsymbol{b})$ | [m$^2$ g$^{-1}$] | RDG–PFA | |
| | Single-scatter albedo | | $\omega(\boldsymbol{b})$ | [–] | | Pre-computation of soot/Legendre coefficients |
| | Scattering phase function | | $p(\theta, \boldsymbol{b})$ | [sr$^{-1}$] | | |
| | Soot/Legendre coefficients | | $\Phi_l(\boldsymbol{b})$ | [–] | Eq. (10) | |
| Plume loading | Observed plume transmittance | | $\tau_{obs}$ | [–] | – – – | Independent |
| Camera pointing | Relative solar azimuth | | $\alpha_s$ | [°] | – – – | |
| | Camera inclination | | $\beta$ | [°] | | Independent |
| | Solar zenith | | $Z_s$ | [°] | | |

[a] The means by which variables are randomly drawn or computed.

[b] The method by which a variable is considered in the GUA MC method. Given a set of "independent" and "random" variables, pre-computed data ($\Psi_{l,sky}(\alpha_s, \beta, Z_s, a_k)$, $\Psi_{l,sun}(\alpha_s, \beta, Z_s, a_k)$, $\sigma_m^e(\boldsymbol{b})$, and $\Phi_l(\boldsymbol{b})$) are obtained and used to calculate soot mass column density.

[c] Sky model coefficients are independent variables in the analysis of a single skylight intensity model ($a \in \{1...15\}$), but, for the derived sky categories ($a \in \{A...D\}$), skylight intensity models are randomized following a discrete uniform distribution with support over the skylight intensity models included in the sky category.

[d] Typical values as listed for each CIE sky model as per the literature (Darula and Kittler, 2002; Kittler et al., 2012).

[e] Author-selected distributions: $\mathcal{U}(0.00, 1.25)$ for CIE sky types 1–6 and $\mathcal{U}(0.75, 1.25)$ for CIE sky types 7–15.

[f] Prior probability distributions for soot properties were derived by Johnson et al. (2013).

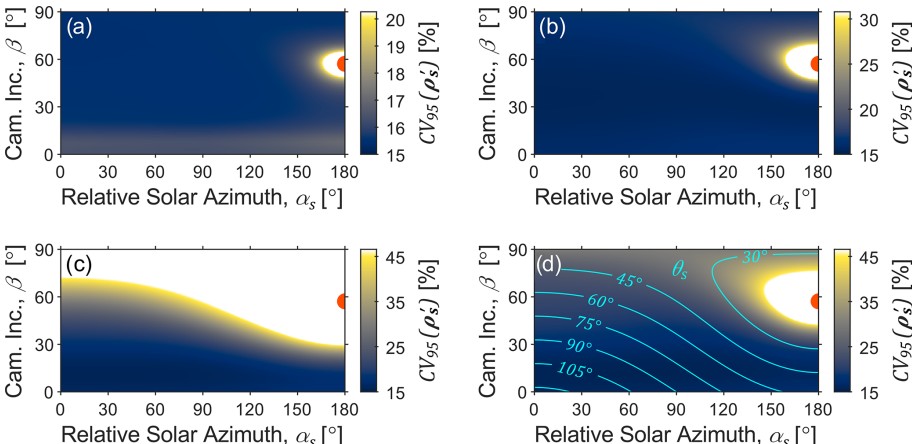

**Figure 3.** Example MC results at solar noon on the summer solstice in Fort St. John, British Columbia – solar zenith ($Z_s$) of 32.8° – for a plume of 90 % observed plume transmittance ($\tau_{obs}$). Panels **(a–d)** show the relative uncertainty ($CV_{95}$) of soot mass column density as a function of relative solar azimuth ($\alpha_s$) and camera inclination ($\beta$) for sky categories A–D, respectively. Contour lines in panel **(d)** show the scattering angle of sunlight into the sky-LOSA camera ($\theta_s$).

alone due to high turbidity that strongly attenuates direct solar radiation, while, for sky category C, the sun's inscattering contribution ($S_{1,sun}$) is significant due to the low turbidity of the sky. Thus, the results for sky category A largely represent the effect of $\theta_s$ on uncertainty through $S_{1,sky}$ while the results for sky category C include an additional (and the most extreme) effect through $S_{1,sun}$. These observations imply that a constraint on $\theta_s$ is necessary and that a stricter constraint is required for lower-turbidity skies.

The mechanism for the decrease in relative uncertainty with increasing $\theta_s$ stems from the optical characteristics of soot particulate. Figure 4 shows statistics of the "energy distribution function" (EDF $= \omega(\boldsymbol{b}) p(\theta, \boldsymbol{b})/4\pi$) discussed by Conrad et al. (2020b), which describes how soot particulate directionally scatters light on an energy basis. The median and $CV_{95}$ of the EDF as a function of angle $\theta$ are shown for the range of the MC-sampled soot properties ($\boldsymbol{b}$) used in the GUA. This is useful for the present discussion since the EDF contains all soot-dependent variables in the computation of $S_{1,sun}$ (see Eq. 7). The figure shows that the EDF, and its influence on $S_{1,sun}$, is much larger and much more uncertain as $\theta_s$ decreases. The influence of these statistics on column density uncertainty depends on the relative intensity of sunlight – specifically $E_{sn}(Z_s, a)/I^o(\alpha_s, \beta, Z_s, a)$ in Eq. (7). If this value is small (highly turbid skies), then variability in the EDF is less important, but if this value is large (low turbidity skies), then variability in the EDF can dominate relative uncertainty in soot mass column density. Interestingly, relative uncertainty in the EDF approaches a constant value towards 90°. This implies that, regardless of the ground-level intensity of the sun, the uncertainty of $S_{1,sun}$ becomes minimal at $\theta_s \gtrsim 90°$ and is within 1 % of this minimum for $\theta_s \gtrsim 60°$ as indicated by the red line in Fig. 4.

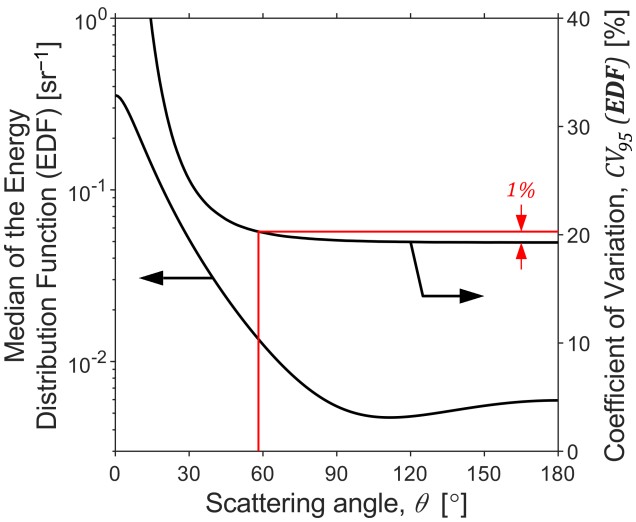

**Figure 4.** Central tendency (median; left logarithmic axis) and relative uncertainty ($CV_{95}$; right linear axis) of the "energy distribution function" (EDF $= \omega(\boldsymbol{b}) p(\theta, \boldsymbol{b})/4\pi$) that dictates the fraction of incident light energy scattered through angle $\theta$ by soot. The magnitude and uncertainty in the EDF are much larger in the forward scattering direction (small $\theta$). As the scattering direction exceeds $\sim 90°$ however, the relative uncertainty in the EDF approaches a constant minimum, implying that the relative uncertainty in $S_{1,sun}$ is minimized for $\theta_s \gtrsim 90°$. The red line shows that relative uncertainty is within 1 % of the minimum if $\theta_s \gtrsim 60°$.

The second trend in Fig. 3 that is generally seen across all measurement conditions is the sensitivity of relative uncertainty to the camera inclination angle ($\beta$). Referring to Fig. 3, much of the observed trend in $\beta$ can likely be attributed to the effect of $\theta_s$, since $\theta_s = \theta_s(\alpha_s, \beta, Z_s)$. However, $\beta$ still influences measurement uncertainty as can be observed in the re-

gion where the $\theta_s$ effect is small ($\theta_s \gtrsim 60°$). Figure 5a through e show example trends in soot mass column density uncertainty as a function of $\beta$ for plume transmittances of 0.25 to 0.95 and a fixed solar azimuth of $\alpha_s = 60°$. These panels each plot $CV_{95}$ as a function of $\beta$, averaged over the range of $Z_s$ that ensures $\theta_s > 60°$. The trends are different for each sky model. Uncertainties for sky categories A and B tend to decrease as the camera inclines, while uncertainties for sky categories C and D increase. The severity of these trends increases with plume transmittance (effectively a reduction in the measured signal), and, as plume transmittance increases towards unity, local minima–maxima in uncertainties as a function of camera inclination may appear. The differing influence of $\beta$ between overcast/partly cloudy and clear-sky models is largely due to the specific CIE sky models, which dictate the gradient in skylight intensity near the horizon and the sun. For clear, low-turbidity skies, intensity gradients are large such that small changes in camera pointing can yield significant changes in inscattered light. By contrast, for higher-turbidity skies, gradients are dampened, and the effect of camera pointing is small.

Figure 5 also shows that the $CV_{95}$ of soot mass column density for sky category C tends to upper bound that of sky category D. This observation holds under most combinations of the MC-independent variables and is a result of the somewhat extreme nature of CIE sky model 12 (lowest turbidity/cleanest atmosphere), which is excluded in sky category D. When considered in sky category C however, CIE sky model 12 tends to increase both the variability and central tendency of soot mass column density, but the relative change in the former is larger – hence, inclusion of CIE sky model 12 typically increases relative uncertainty. This implies that sky category C reliably imposes the largest constraints on camera positioning and can therefore be used to conservatively locate sky-LOSA equipment under clear skies. However, if in a dense industrial area, where the clearest-sky model is not relevant, the less-constraining sky category D can instead be used as noted in Sect. 3.1.2.

### 4.1.1 Camera pointing heuristics

Upon arrival at a measurement facility, the sky-LOSA user's first task is to determine the position of the sky-LOSA camera for data acquisition. This important decision can be made by considering viable camera pointings from GUA MC data through constraints on $\beta$ and $\theta_s$. That is, for a user-identified sky category and plume transmittance and given the position of the sun, the sky-LOSA camera's pointing can be constrained based on a desired threshold in a relative uncertainty, $CV_{95}$. Table 3 provides an example of such constraints. The table lists bounds on the camera inclination angle ($\beta$) and solar scattering angle ($\theta_s$) for each sky category, given the plume transmittance: where bounds were computed by determining where $CV_{95} \leq 17\%$ for all values of $Z_s \in [0, 90°]$, such that the results are independent of solar zenith.

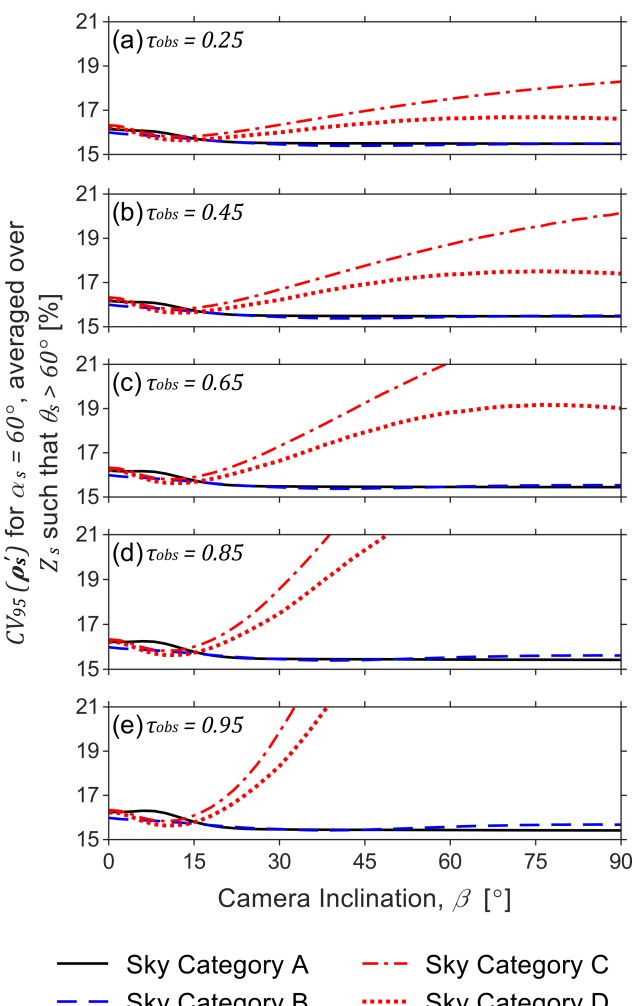

**Figure 5.** Percentage relative uncertainty in soot mass column density for a relative solar azimuth ($\alpha_s$) of 60° averaged over all solar zenith angles ($Z_s$) as a function of camera inclination angle ($\beta$). Data are plotted for each sky category for observed transmittances ($\tau_{obs}$) of 0.25 to 0.95 in panels (**a–e**). For sky categories A and B (representing overcast and partly cloudy skies), uncertainty can be *slightly* reduced by ensuring the camera inclination angle exceeds approximately 15°. For sky categories C and D (representing clear skies), minimal uncertainty is achieved at camera inclination of 9.75–12.50°, and uncertainty can drastically increase for camera inclination increase beyond 20° – however, the effect becomes muted for more optically thick plumes (lower $\tau_{obs}$). These observations support the general heuristic for clear skies that the camera inclination angle should kept below $\sim 20°$, especially for lightly and moderately sooting flares.

One additional consideration in the pointing of the sky-LOSA camera is the direction of plume propagation. Under quiescent conditions, buoyancy-driven flare plumes will propagate vertically away from the flare stack; however, under sufficiently strong crosswinds, the flame and plume can bend over and propagate horizontally, parallel to the wind di-

**Table 3.** Summary of constraints regarding camera pointing relative to the horizon and sun as a function of plume transmittance ($\tau_{obs}$). Compliance with these heuristics ensures that uncertainty in sky-LOSA-computed soot mass column density is low (CV$_{95} \leq 17\%$), regardless of solar zenith angle ($Z_s$). (N. C. denotes "no constraint").

| Sky category | Camera inclination angle, $\beta$ | Solar scattering angle, $\theta_s$ |
|---|---|---|
| Plume transmittance ($\tau_{obs}$) $\leq 0.45$ | | |
| A | N. C. | $\theta_s \geq 13°$ |
| B | N. C. | $\theta_s \geq 23°$ |
| C | $\beta \leq 20°$ | $\theta_s \geq 79°$ |
| D | $\beta \leq 25°$ | $\theta_s \geq 54°$ |
| Plume transmittance ($\tau_{obs}$) $\leq 0.70$ | | |
| A | N. C. | $\theta_s \geq 15°$ |
| B | N. C. | $\theta_s \geq 29°$ |
| C | $\beta \leq 16°$ | $\theta_s \geq 83°$ |
| D | $\beta \leq 19°$ | $\theta_s \geq 59°$ |
| Plume transmittance ($\tau_{obs}$) $> 0.70$ | | |
| A | $\beta \geq 10°$ | $\theta_s \geq 17°$ |
| B | N. C. | $\theta_s \geq 35°$ |
| C | $\beta \leq 13°$ | $\theta_s \geq 84°$ |
| D | $\beta \leq 17°$ | $\theta_s \geq 62°$ |

rection. In this latter case, if the plume propagates towards or away from the sky-LOSA camera, turbulent plume structures of differing vorticity become overlapped from the camera's perspective. Therefore, it is best to position the sky-LOSA camera such that it points orthogonally to the wind direction, which minimizes out-of-image plane motion of the plume and yields the best data for velocimetry calculations. This should be viewed as a weak constraint on sky-LOSA data acquisition however, since the effect of uncertainty in estimated velocity on mass emission rate is generally negligible compared to that of column density uncertainty.

### 4.1.2 Further camera heuristics

Following selection of a permissible sky-LOSA camera position, the imaging optics must be chosen. Prime (fixed focal length) lenses are employed in the sky-LOSA technique to avoid ambiguity in optical magnification and, hence, spatial scaling of the image. The most appropriate prime lens for the studied flare is one that maintains the entirety of the flare flame well within the image during the data acquisition period. This helps to ensure that a control surface within the image plane that transects the plume and encloses the flame can be derived, as shown in Fig. 1. For a flame that is relatively unsteady – i.e., moving with the wind – it is suggested to keep the flare flame approximately one-quarter of the smallest image dimension. By contrast, if the flame

is steady, a flame length of approximately one-third of the smallest image dimension should be targeted. For the sky-LOSA camera used by the authors (minimum sensor dimension of $\sim 14$ mm), a good rule of thumb is that the appropriate focal length ($f$, [mm]) will be on the order of

$$f\,[\text{mm}] \approx \frac{4.1}{\cos\beta}\frac{H}{L_f}, \tag{16}$$

where $H$ is the horizontal stand-off distance from the flare stack [m], and $L_f$ is the length of the flare flame [m]. However, use of a prime lens necessitates a trade-off between lens focal length, horizontal stand-off distance, and size of the flame within the image.

With an appropriate lens selected, the user must then choose imaging parameters that influence the exposure and focus of the image. The objective is to obtain an image that maximizes the digital signal while minimizing exposure time and ensuring the flame is in focus. In the authors' experience, this can be obtained with a lens aperture close to full-open (typically $f$-number $\leq 5.6$) and an exposure time less than $\sim 2$ ms. Prior to acquiring the image data, the flame and flare stack should be brought into focus. The user can then obtain sky-LOSA data for the desired duration; it is recommended, however, that a minimum dataset of 10 min be obtained to permit good convergence of the time-averaged soot emission rate.

### 4.2 Open-source software tool for simpler sky-LOSA setup – SetupSkyLOSA

While the camera pointing heuristics presented in Table 3 can be used to ensure that CV$_{95} \leq 17\%$ for the listed plume transmittances regardless of solar zenith angle, this simplified set of constraints is also necessarily overly conservative and excludes specific combinations of inputs that might produce similar or better uncertainties in different scenarios. An even better approach would be to use the wealth of computed GUA data to provide camera position and pointing constraints on a case-by-case basis. This is made possible using a new open-source software tool, SetupSkyLOSA (Conrad, 2020), that was developed as part of this work using the presented GUA MC data. This MATLAB-based application enables a sky-LOSA user to probe statistics of soot mass column density (such as CV$_{95}$) for their specific measurement conditions. The key output of the software tool is an image of the desired soot mass column density statistic plotted as a function of absolute camera pointing. The software tool is briefly described in this section and employed in a case study in Sect. 4.3.

Figure 6 shows a flow chart describing the SetupSky-LOSA software's main procedure. For a user-inputted location and time, the software first determines the current position of the sun using an integrated solar position calculator – a MATLAB implementation of the National Renewable Energy Laboratory's (NREL's) Solar Position Algorithm (SPA)

(Reda and Andreas, 2008). The SPA returns the solar zenith ($Z_s$) and absolute bearing of the sun ($\alpha_{sN}$, where the subscript "N" implies the absolute bearing measured clockwise from *true* north) at the current time and over the measurement date. The user also inputs the index ($a$) of the most appropriate CIE sky model or category, an estimate of the observed plume transmittance at the sky-LOSA measurement wavelength ($\tau_{obs}$), and the desired statistic of soot mass column density ($\eta$). With these inputs, SetupSkyLOSA then loads the GUA MC data for the selected sky model or category and interpolates for the desired statistic using the current solar zenith and estimated plume transmittance.

At this point, the software has computed $\eta(\alpha_s, \beta)$ for the user's current set of independent variables ($Z_s, \tau_{obs}, a$). However, rather than plotting $\eta(\alpha_s, \beta)$ – i.e., using the relative bearing – the software uses the known absolute bearing of the sun ($\alpha_{sN}$, computed by the SPA) to plot $\eta(\alpha_N, \beta)$. That is, the requested statistic is plotted as a function of *absolute* camera bearing and inclination, which together define the camera pointing. The user can then easily determine an acceptable camera pointing using a laser rangefinder (for inclination) and a compass (corrected to true north), laser rangefinder, or GPS device (for absolute bearing).

SetupSkyLOSA also includes several added utilities to support optimal positioning and pointing of the sky-LOSA camera. Firstly, using the same pinhole camera model that enables spatial scaling of the image, the software tool can optionally overlay the approximate extent of the image sensor in the $\alpha_N - \beta$ domain, based on sensor dimensions, employed optics, and the pointing of the centre of the image. This helps a user ensure that the entirety of the image frame – including the eventual control surface used for emission rate calculation – has reasonable levels of measurement uncertainty, given a user-selected lens of known focal length. To support ideal velocity calculation, the software can also overlay camera pointings ($\alpha_N$) that are closely orthogonal to the wind. This follows the heuristic discussed in Sect. 4.1.1. The software shows the optimal range of camera pointing as orthogonal to the wind $\pm 18.2°$, which corresponds to 5 % out-of-image plane motion ($\cos^{-1}(0.05)$) TS3. Two additional utilities are not shown in Fig. 6. The "maximizer" utility computes the maximum of a chosen relative uncertainty statistic over a user-defined period. This tool allows a user to seek camera pointings that yield satisfactory uncertainties as the sun moves during the anticipated duration of the sky-LOSA measurement. The "positioner" utility takes the plotted relative uncertainty statistic as a function of camera *pointing* and provides region(s) where the sky-LOSA camera may be *positioned* relative to the flare stack given the flare stack height, maximum horizontal stand-off distance from the flare, and relative uncertainty threshold. The user can optionally print these permissible regions to a .kml file for use in mapping software like Google Earth. These latter two utilities are employed in the following case study.

## 4.3 Case study – Atasta facility

The utility of the novel software tool, SetupSkyLOSA, is shown in this section via a case study of a sky-LOSA measurement at a real oil and gas facility. The Atasta Gas Processing and Transport Centre (Centro de Proceso y Transporte de Gas Atasta) is a midstream oil and gas facility near Atasta, in the Mexican state of Campeche. The facility is under the jurisdiction of Petróleos Mexicanos and receives sour gas and condensates from the Cantarell offshore oil field for processing and transport to the national market. As shown in Fig. 7a, the Atasta facility is located 35 km west of Ciudad (Cd) del Carmen and approximately 5 km south of the shore of the Bay of Campeche. The facility occupies approximately 1 km$^2$, with most infrastructure in the southeast corner of the site as visible in Fig. 7b. Flaring activities include multiple pit-style and vertical stack flares, which are in the northwest corner of the site.

For this case study, a sky-LOSA measurement of soot emissions from the central flare stack at the Atasta station was considered as indicated in Fig. 7b, the base of which is located at $18°38'41.46''$ N and $92°10'08.59''$ W. The following example measurement details are assumed.

1. The sky-LOSA measurements occur on 13 May 2021, which is the date of that year that the sun most closely reaches the solar zenith ($Z_s = 0°$).

2. Predicted sky conditions are uncertain and may change between overcast and fully clear conditions throughout the day.

3. Wind speed is predicted to be low and the flare is strongly buoyant.

4. The flare stack is 30 m in height, and the horizontal stand-off distance of the sky-LOSA camera is limited to 250 m or less due to available optics.

5. The flare is lightly sooting, with an observed transmittance of approximately 90 % ($\tau_{obs} \approx 0.90$).

Assumption 2 implies that sky-LOSA data acquisition may occur under skies represented by any of sky categories A–D. Furthermore, assumption 3 suggests that the soot-laden flare plume propagates vertically from the flare stack, and, therefore, the constraint on camera position with respect to wind direction is unimportant. The sky-LOSA user wishes to obtain sky-LOSA data with minimal measurement uncertainty, while also avoiding re-location of the sky-LOSA camera throughout the day, if possible.

Given the known GPS coordinates of the flare stack, measurement date, and approximate plume transmittance, SetupSkyLOSA can be used to constrain sky-LOSA camera pointing for any sky condition and time of the day based on the $CV_{95}$ of soot mass column density. Since the user wishes to avoid re-location of the sky-LOSA camera, camera position

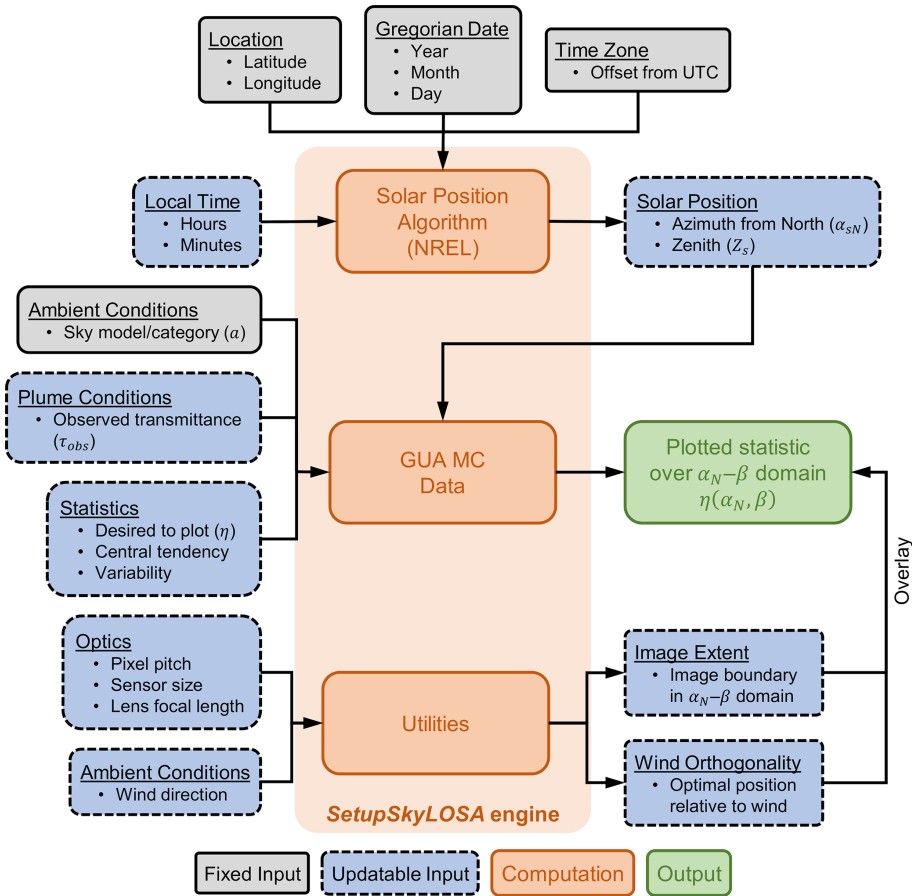

**Figure 6.** Flow chart of the main procedure of the SetupSkyLOSA software tool. The user provides the location, Gregorian date, time zone, and local time, which are used to compute the corresponding solar position using the Solar Position Algorithm of the National Renewable Energy Laboratory (NREL; Reda and Andreas, 2008). Then, with data on ambient conditions and observed plume transmittance, the software tool plots the desired statistic of soot mass column density over the $\alpha_N - \beta$ domain. Additional utilities include the overlay of the image sensor and optimal positioning relative to the wind on the plotted statistic, in addition to the "maximizer" and "positioner" utilities, which are not shown in the figure. The latter two utilities permit a user to identify acceptable camera positions and pointings over a measurement period and output these data in .kml format for use with mapping software such as Google Earth.

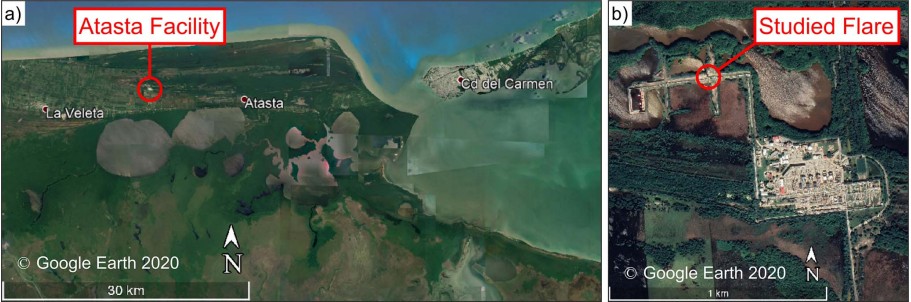

**Figure 7. (a)** Location of the Atasta Gas Processing and Transport Centre in the Mexican state of Campeche, 35 km west of Ciudad (Cd) del Carmen and approximately 5 km south of the Bay of Campeche. **(b)** Location of the flare that is the focus of the case study located in the northwest corner of the site amongst other flaring infrastructure.

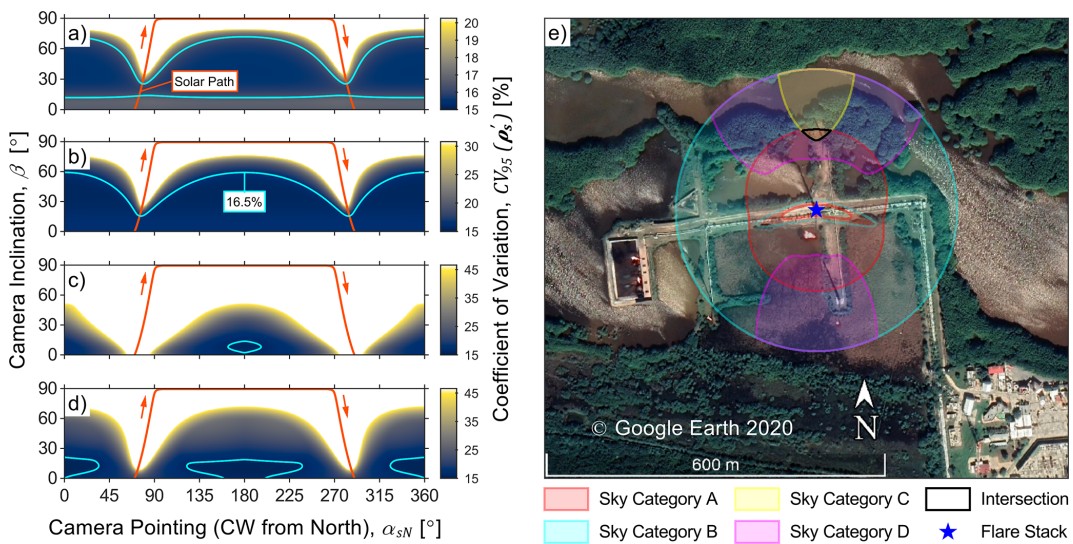

**Figure 8. (a–d)** Relative uncertainty ($CV_{95}$) in soot mass column density as a function of camera inclination ($\beta$) and pointing ($\alpha_{sN}$), maximized over the measurement day for sky categories A–D, respectively, given an observed plume transmittance ($\tau_{obs}$) of 0.90. Overlaid in the figures is the path of the sun over the day (which approximately reaches $Z_s = 0°$ at 13:05 central daylight time) in addition to a contour of $CV_{95} = 165\%$. **(e)** Permissible regions for sky-LOSA camera positioning relative to the studied flare stack for sky categories A–D. The black line shows the intersection of these regions which is $\sim 600\,m^2$ at which the sky-LOSA camera can be positioned to minimize measurement uncertainty, regardless of sky conditions.

and pointing should be constrained using the relative uncertainty maximized over the day. The "maximizer" utility of SetupSkyLOSA permits this calculation for each of the sky categories; results are shown in Fig. 8a–d for sky categories A–D. Noting the differing colour scales in the four figures, there is significant variability in sky-LOSA uncertainties for each of the sky categories, as in Fig. 3. For further context, the path of the sun over the measurement date is overlaid in Fig. 8a–d in addition to a contour of $CV_{95} = 16.5\%$, which is within 1 % of the best attainable uncertainty for these conditions.

Using the uncertainty data in Fig. 8a–d, the positioner utility of SetupSkyLOSA can be employed to highlight where the sky-LOSA camera may be positioned relative to the flare stack. This was performed for each of the sky categories using the uncertainty threshold of 16.5 %. Permissible camera positions were output in .kml format by the positioner CE3 utility and are overlaid on a map of the Atasta facility in Fig. 8e and are quite different for each of the sky categories. Permissible camera positions for sky category B exist beyond a small region near the stack tip, while those for sky category A are within an annular region surrounding the flare stack – since the lower limit on the camera inclination angle in Fig. 8a imposes a maximum permissible stand-off distance. Sky category D contains two permissible regions – one to the south and one to the north of the flare stack – while the most-constrained sky category C has one relatively small region to the north of the flare stack. Recalling assumption 2 that predicted sky conditions were uncertain, the sky-LOSA

user should ideally position the camera at the intersection of the sky-category-dependent permissible regions. This small area is outlined in black in the figure, $\sim 136\,m$ due north of the flare stack and just $604\,m^2$ in size ($\sim 0.31\%$ of the 250 m radius region). It is apparent in Fig. 8e that this ideal position intersects a clearing in the treed area where the sky-LOSA camera should be positioned for the specific conditions of this case study.

This case study shows the remarkable utility of the Setup-SkyLOSA software tool. The tool quickly provides resolved measurement uncertainty data from the GUA that would otherwise require millions of MC analyses to compute. These uncertainty data enable optimal sky-LOSA camera positioning and pointing and also represent a first-order estimate of soot emission rate uncertainties that are computed in postprocessing. Together with the additional utilities and general camera heuristics, this software tool permits a sky-LOSA user to obtain optimal sky-LOSA data that minimize measurement uncertainties under generalized conditions.

## 5 Conclusions

A comprehensive Monte Carlo-based general uncertainty analysis (GUA) has been used to develop heuristics constraining the pointing and positioning of sky-LOSA equipment for measurement of soot–black carbon emissions from gas flares. The GUA identifies generalized constraints based on predicted measurement uncertainties in soot mass column density, computed using sky-LOSA. The results show that

equipment setup constraints can be classified based on the conditions of the sky, relative positioning of the sun, and inclination angle of the camera. With additional heuristics on camera optics and imaging parameters, the presented re-

5 sults provide generalized guidance to greatly simplify acquisition of optimal sky-LOSA data in the context of complex, non-linear measurement uncertainties. These are further extended in the open-source software utility, SetupSkyLOSA, which interprets the GUA results to provide detailed guid-

10 ance for any specific combination of location, date–time, and flare, plume, and ambient conditions. Furthermore, software-displayed soot mass column density statistics provide the user with a first-order estimate of measurement uncertainty in soot–black carbon emission rate that otherwise is only com-

15 putable during post-processing. The case study using Setup-SkyLOSA to identify optimal equipment setup at a real oil and gas facility in Mexico demonstrates the utility of this new software tool, which as an open-source application can hopefully facilitate broader use of the sky-LOSA technique and

20 ultimately help increase the knowledge base of soot–black carbon emissions from gas flares.

## Appendix A: Implementation details of the updated MC method

### A1  Truncation of the expanded SPF

For TS4 the prior probability distributions of soot properties derived by Johnson et al. (2013), the total order of soot coefficients required to represent the soot SPF according to the procedure of Schuster (2004) was typically $L(\boldsymbol{b}) = 76$ (median). In the most extreme case however, representing a strongly forward scattering soot population (corresponding to large soot aggregate size), $L(\boldsymbol{b})$ reached 698, suggesting that pre-computation of the sky and sun coefficients up to $\Psi_{698}$ would be necessary to compute the inscattering correction in the worst case. While calculation of the sun coefficients to this large order is generally trivial, calculation of the sky coefficients becomes overwhelmingly cumbersome as the order increases. Importantly though, like the soot coefficients, the magnitude of the sky coefficients approaches zero as the order approaches infinity; therefore, the product of $\Phi_l(\boldsymbol{b})$ and $\Psi_{l,\mathrm{sky}}(\alpha_s, \beta, Z_s, \boldsymbol{a})$ more rapidly decreases in magnitude as $l$ increases than either component alone. This permits further truncation of the series for the calculation of $S_{1,\mathrm{sky}}$. Specifically, calculation of $S_{1,\mathrm{sky}}$ via Eq. (11) using $L(\boldsymbol{b}) = 200$ was consistently in close agreement with direct numerical integration via Eq. (7) – where, over $10^6$ randomized sets of $(\alpha_s, \beta, Z_s, a, \boldsymbol{b})$, the median relative difference was just $2.3 \times 10^{-7}$. This implies that it is acceptable to impose that $\Psi_{l,\mathrm{sky}}(\alpha_s, \beta, Z_s, a) = 0, \forall l > 200$.

### A2  Incident intensity-normalized solar normal irradiance

As noted in Sect. 2, ground-level solar normal irradiance – $E_{\mathrm{sn}}(Z_s, a)$, usually measured via solar images taken in the field – can be modelled using the CIE skylight models. To accomplish this, the incident intensity-normalized solar normal irradiance is expanded:

$$\frac{E_{\mathrm{sn}}(Z_s, a)}{I^\circ(\alpha_s, \beta, Z_s, a)} = \frac{E_{\mathrm{sn}}(Z_s, a)}{D_{\mathrm{h}}(Z_s, a)} \frac{D_{\mathrm{h}}(Z_s, a)}{I^\circ(\alpha_s, \beta, Z_s, a)}, \quad \text{(A1)}$$

where $D_{\mathrm{h}}(Z_s, a)$ is the diffuse horizontal irradiance, calculable for the CIE models via numerical integration of

$$D_{\mathrm{h}}(Z_s, a) = \int_0^{2\pi} \int_0^{\frac{\pi}{2}} I(\alpha, \alpha_s, Z, Z_s, a) \cos Z \sin Z \, \mathrm{d}Z \, \mathrm{d}\alpha, \quad \text{(A2)}$$

which is independent of the value of $\alpha_s$.

The ratio of solar normal to diffuse horizontal irradiance is complex to quantify in a general sense as it is a function of atmospheric composition. However, for the purposes of the present GUA it is modelled as follows:

$$\frac{E_{\mathrm{sn}}(Z_s, a)}{D_{\mathrm{h}}(Z_s, a)} = \frac{E_{\mathrm{sh},o}(Z_s, a) \exp\left(-m(Z_s)\, \sigma^{\mathrm{e}*}(m)\, T(a)\right)}{D_{\mathrm{h}}(Z_s, a) \cos Z_s}. \quad \text{(A3)}$$

The numerator of the righthand side is the ground-level solar horizontal irradiance, calculated as the product of the extraterrestrial (subscript "$o$") solar horizontal irradiance ($E_{\mathrm{sh},o}(Z_s, a)$) TS5 with an exponential representing attenuation through the atmosphere. In computation of the latter, $m(Z_s)$ is the relative air mass quantifying the amount of air in the atmosphere at the solar zenith angle relative to the vertical direction, $\sigma^{\mathrm{e}*}(m)$ is the ideal extinction for a clean atmosphere at a given relative air mass, and $T(a)$ is the model-dependent turbidity factor that is used to consider realistic atmospheres and describes the number of clean atmospheres required to represent attenuation through the non-ideal, polluted atmosphere. In the denominator, the cosine is included to transform the numerator into the required ground-level solar *normal* irradiance. In the present work, typical values of the turbidity factor ($T(a)$) and irradiance ratio $D_{\mathrm{h}}(Z_s, a)/E_{\mathrm{sh},o}(Z_s, a)$ are taken from Darula and Kittler (2002) and Kittler et al. (2012), while their recommended formulations for $m(Z_s)$ (Kasten and Young, 1989) and $\sigma^{\mathrm{e}*}(m)$ (Navvab et al., 1984) are employed:

$$m(Z_s) = \left(\cos Z_s + 0.50572(96.07995° - Z_s)^{-1.6364}\right)^{-1}$$
$$\sigma^{\mathrm{e}*}(m) = (9.9 + 0.043m)^{-1}. \quad \text{(A4)}$$

To model some amount of unknown uncertainty due to the use of "typical" metrics listed in the literature, an additional randomized variable ($\xi_{\mathrm{ED}}(a)$) was introduced as a scalar multiplier to the sun component of the inscattering correction. For CIE sky models with an unobstructed sun (types 7–15), $\xi_{\mathrm{ED}}(a) \sim \mathcal{U}(0.75, 1.25)$, and for models with an obstructed sun (types 1–6), $\xi_{\mathrm{ED}}(a) \sim \mathcal{U}(0, 1.25)$. These prior distributions of $\xi_{\mathrm{ED}}(a)$ were based on observations by Watanabe et al. (2016), who studied the "clearness index" (ground-level horizontal normalized by extraterrestrial horizontal irradiance) over 5 years at 47 observation stations across Japan. They found that the relative variation in the clearness index was approximately 4.3 % for skies with unobscured suns and 35 % for skies with obscured suns; corresponding variance-equivalent uniform distributions would have a range of 15 % and 121 %, respectively. For skies with an unobscured sun, this range was expanded to 50 % (0.75–1.25) to give a conservatively broad prior since measurement uncertainty can be quite sensitive to solar irradiance. By contrast, for skies with an obscured sun, where the solar irradiance has a small contribution to measurement uncertainty, the uniform distribution was only slightly widened to 125 % (0–1.25).

*Data availability.* The developed software tool and associated data are available online as open-source and build distributions (https://doi.org/10.5281/zenodo.3908540; Conrad, 2020).

*Author contributions.* Both authors conceptualized the research and developed the methodology. MRJ was responsible for funding acquisition, project administration, provision of resources, and supervision. BMC curated the data, performed the formal analysis and investigation, developed the software, and produced the original manuscript including visualizations. Both authors reviewed and edited the manuscript throughout the publication process.

*Competing interests.* The authors declare that they have no conflict of interest.

*Acknowledgements.* We are grateful for the support of Michael Layer (project manager, Natural Resources Canada) for championing this and related projects and to Brian Crosland (Natural Resources Canada) for lending computational resources.

*Financial support.* This research has been supported by Natural Resources Canada (grant no. CH-GHG IETS-19-103) and the Natural Sciences and Engineering Research Council of Canada (NSERC, grant nos. 479641, 06632, 522658). CE4

*Review statement.* This paper was edited by Oleg Dubovik and reviewed by three anonymous referees.

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

**Remarks from the language copy-editor**

CE1 Please note it is our house standard to denote relationships, coupling, etc., between two or more things as well as "and" with an en dash. A slash technically means "and/or" and shouldn't be used here. An en dash more appropriately fits the meaning here and has already been made consistent throughout.

CE2 Please give an explanation of why this needs to be changed. We have to ask the handling editor for approval. Thanks.

CE3 Please note it is our house standard to only format terms in quotation marks at their instance. Our apologies that this was overlooked.

CE4 Please note the slight edit for grammatical correctness.

**Remarks from the typesetter**

TS1 Please ask Matthew R. Johnson to log into his account at Copernicus and add the ORCID number. Thank you.

TS2 In ranges and series, it is our house standard to retain only the final unit of measure. This is in line with the ACS Style Guide (p. 226). If this rule leaves room for misinterpretation in the context of your manuscript, the unit can be repeated. Please advise us on this matter.

TS3 Please give an explanation of why this needs to be changed. We have to ask the handling editor for approval. Thanks.

TS4 Please note that I removed the dots in e.g. "Eq. A1" to be consistent in this paper. Thank you for pointing this out.

TS5 Thank you for pointing this out. I changed the formatting to be consistent.

TS6 Please provide date of last access.

TS7 Please provide date of the conference.

TS8 Please provide date of last access. Please note that the issue no. is not needed.

TS9 Please provide date of last access.