# Peer review of "An uncertainty-based protocol for the setup and measurement of soot/black carbon emissions from gas flares using sky-LOSA"

_Atmospheric Measurement Techniques, 2020_

## Short Comment (SC1) · 11 Nov 2020

I am a government researcher studying air emissions from the oil and gas industry and am in the process of purchasing equipment and building the knowledge and skills to perform skyLOSA measurements both in Canada and abroad. I am eager to use the setupSkyLOSA tool to see if the locations we chose while acquiring flare images during fieldwork in 2019 were well-suited for skyLOSA.

I especially appreciate the detailed description of the uncertainties in the paper. Thorough and open presentation of model and measurement uncertainties is critical when employing complex tools in the context of measurement, reporting, and verification of emissions. In the MRC context, the setupSkyLOSA tool without the accompanying peer-reviewed publication would lose much, if not all of its usefulness.

I also see great value in the numerous rules of thumb this manuscript provides, which enable a potential sky-LOSA setup operator to set up their equipment and acquire images faster and with greater confidence. This is especially important in the field where changing wind direction, sun location, and clouds can quickly render a once-suitable location unusable.

---

## Referee Comment (RC1) · Anonymous Referee #3 · 17 Nov 2020

This manuscript presented an uncertainty-based guide/instruction for potential users of the sky-LODA technique to measure soot/black carbon emissions from flares in the oil and gas industry. Although the method itself and the details of various models involved in the data analysis have been published by the authors, the paper is useful to help potential researchers/engineers to better understand and apply this promising technique. The paper is well written as deserves publication. I have few questions/comments for the authors to consider. 1. In line 29 on page 4, please explicitly list the eight soot properties. 2. In the derivation of Eq. (4), do you need to assume that the mass-normalized

extinction cross section of soot is constant everywhere in the flare? Otherwise, it cannot be taken out of the second integral on the right hand side of Eq. (4). 3. In this work, the authors dealt with the in-scattering term by expanding the SPF in Fourier-Legendre series. As the authors admitted, this expansion requires a large number of terms to have an accurate representation of the SPF, especially when it is highly-forward peaked. There are other methods to more efficiently deal with highly-forward peaked SPF, such as the Henyey-Greenstein and transport approximations, have the authors considered such approaches in the calculations of the in-scattering terms? 4. In this paper, the authors did not provide the detailed soot properties, but reference to a previous study. It appears that the authors assume that the soot properties are uniform over the entire flare under consideration and also remain the same from one flare to another. Is this correct? If yes, the authors need to justify this assumption. The soot properties in a flare with soot emission reduction measure (such as partial premixing) may not be the same as those without any such measures. 5. The choice of sky model group seems not straightforward and has a strong influence on the measurement. I wonder if the authors can provide more useful 'tips' to make such a choice for new users of sky-LOSA.

---

## Referee Comment (RC2) · Anonymous Referee #1 · 21 Nov 2020

The manuscript amt-2020-255 by Conrad and Johnson describes a software heuristic for assisting a remote optical technique (skylight line-of-sight attenuation; skyLOSA) for measuring soot/black carbon emissions in large industrial flames. The technique allows a user to select the most reasonable position to set up the skyLOSA camera for a given set of flare and sky conditions. The computations behind this technique are intensive, so the manuscript spends some time describing a useful pre-computation approach. The pre-computed values are later used as inputs to a Monte Carlo uncertainty calculation.

[Figure]

From the skyLOSA perspective, the manuscript does not present new concepts or results. The main novel concept of this work is to apply the same theory used during detailed analysis to measurement, so that the measurement can be carefully configured to provide optimal results. This is a generally interesting concept, but could be considered as a technical note rather than a manuscript.

I recommend substantially shortening the manuscript's description of skyLOSA in order to reflect the subsequent conclusions. I also have reservations about the assumptions made in the MC analysis, especially the assumption that the flame emits only soot and no volatiles. These and other comments are detailed below.

**Length.** The manuscript often reads like a hybrid between a doctoral thesis and an instrument manual, especially in Sections 2, 3, 4.1.2, and 4.2. The text is well written, but inappropriately long. The audience here is not reading to reproduce skyLOSA calculations, but to understand the general concepts used. Please either cite other work or move this text to a supplement. This text can be replaced by short descriptions focussed on key concepts.

Similarly, too many acronyms are used in these sections and are not used frequently enough to be necessary (including ET, SPF, CM-LHS, ...) and not all symbols are defined next to their equations (e.g. L(b) in Equation 9 and $a_k$ next to Equation 18).

**Monte Carlo clarification.** A Monte Carlo calculation randomly samples prior distribution(s) and repeats a calculation in order to obtain a posterior distribution of results. The key question here is what priors were assumed, and how accurate are they? The manuscript glosses over this point and takes the MC output as correct without any top-down validation.

Please revise Section 3 and Table 2 to emphasize the prior distributions used. The authors have already done this in their earlier work (Johnson et al., 2013, Table 2) by tabulating "Distributions used in MC". I believe the authors did intend to include this information but I do not find it clear enough. In Table 2 of this work, the last column "MC

Implementation" is specified as "MC-randomized" multiple times – this is a meaningless statement. Of course the MC calculation performs random sampling.

**Validation of assumptions.**

The manuscript assumes throughout that a perfect skyLOSA measurement gives a perfect result. This has not been justified in the manuscript nor in earlier skyLOSA work, to my knowledge. These calculations are not constrained by any direct measurements. The skyLOSA approach is comparable to a satellite retrieval algorithm and requires direct validation. Until directly validated, this limitation must always be repeated. The concentrations reported by the current approach are a type of "equivalent black carbon" defined by the authors' assumptions.

My main concern here is with respect to the aerosol optical properties, which have not been discussed at all. Instead, Johnson et al. 2013 is cited. The authors have assumed that the flame emits only soot. What about organics, which may condense when the plume cools? The photograph in Johnson et al. 2013 clearly suggests that the plume may have cooled before measurement. What about inorganics such as sulfates? How pure are the fuels burnt in these flares? Any impurities are likely to influence the aerosol optical properties.

Please add calculations where black carbon is assumed to be mixed with organics or other impurities, using reasonable and literature-based assumptions, and show how the conclusions of this work change in response.

A lesser concern is the assumption (Section 3.1.3) of an ideal clean atmosphere. What about background aerosol? Surely the air around an oil field is not perfectly clean.

**Other minor comments.**

The justification of a quantile-based coefficient of variation in Section 3.2.1 can be shortened.

The word compiled in Section 3.1.5 should probably be changed to grouped. And I am

not sure I understand what concept the authors are trying to convey here. Was the grouping done based on skyLOSA results?

I found the discussion of the total order L(b) in Section 3.1.4 unclear. Is this discussion significant, considering the uncertainties in the assumption of a black-carbon-only aerosol and aerosol-free sky?

––––––––––––––––––––––––––––––––––

---

## Author Comment (AC1) · 18 Dec 2020

**Manuscript ID:** amt-2020-255
**Title:** An uncertainty-based protocol for the setup and measurement of soot/black carbon emissions from gas flares using sky-LOSA
**Authors:** Conrad, Bradley M.; Johnson, Matthew R.

**Point-by-point Response to Comments by Referee #1**

*The manuscript amt-2020-255 by Conrad and Johnson describes a software heuristic for assisting a remote optical technique (skylight line-of-sight attenuation; skyLOSA) for measuring soot/black carbon emissions in large industrial flames. The technique allows a user to select the most reasonable position to set up the skyLOSA camera for a given set of flare and sky conditions. The computations behind this technique are intensive, so the manuscript spends some time describing a useful pre-computation approach. The pre-computed values are later used as inputs to a Monte Carlo uncertainty calculation.*

*From the skyLOSA perspective, the manuscript does not present new concepts or results. The main novel concept of this work is to apply the same theory used during detailed analysis to measurement, so that the measurement can be carefully configured to provide optimal results. This is a generally interesting concept, but could be considered as a technical note rather than a manuscript.*

*I recommend substantially shortening the manuscript's description of skyLOSA in order to reflect the subsequent conclusions. I also have reservations about the assumptions made in the MC analysis, especially the assumption that the flame emits only soot and no volatiles. These and other comments are detailed below.*

We thank the Referee for their review of this manuscript and have addressed their comments on a point-by-point basis below. While this manuscript does not present new sky-LOSA *field measurement data*, it does describe novel and significant improvements to the sky-LOSA technique. These noteworthy advancements enable both a thorough general uncertainty analysis and considerably accelerated post-processing of sky-LOSA data, as detailed in our responses below. Moreover, the new open-source software tool developed, tested, and released in conjunction with this manuscript, extends the presented general uncertainty analysis data and, for the first time, allows in-field estimates of uncertainty at the time of measurement, greatly simplifying the use of the sky-LOSA technique and opening it up to others.

Based on the Referee's feedback we have nevertheless made revisions to shorten the manuscript, while further strengthening the discussion and referencing. These include a reduction in the overall length by moving some details of the employed Monte Carlo (MC) method to a new appendix, Appendix A. While the validation of assumptions noted as technical concerns by the reviewer have been largely addressed in previous work, we have made these references clearer in the manuscript and have revised text where necessary to ensure the justification of assumptions in the present work is evident.

**General Comments:**

**Length.** *The manuscript often reads like a hybrid between a doctoral thesis and an instrument manual, especially in Sections 2, 3, 4.1.2, and 4.2. The text is well written, but inappropriately long. The audience here is not reading to reproduce skyLOSA calculations, but to understand the general concepts used. Please either cite other work or move this text to a supplement. This text can be replaced by short descriptions focussed on key concepts.*

The presented methodology – specifically the expansion of the scattering phase function, creation of sky categories/groups, and implementation of the variance-reducing MC method – is a novel and significant advancement to the sky-LOSA algorithm that was necessary to enable the presented general uncertainty analysis. Moreover, this new approach is now used in the post-processing phase of sky-LOSA to more-efficiently compute flare BC emissions from image data. Therefore, to explicitly justify necessary assumptions and to support others in the analysis of sky-LOSA data, we chose to describe the procedure in detail. However, we do agree that some of this theory can be placed in an appendix without sacrificing the readability of the manuscript and the understandability of the results and implications.

To reduce the length of the main manuscript, we have made the following revisions:

- Merged original Sections 3.1.1 and 3.1.2 into a single section (Section 3.1.1 in the revised document) that describes the Fourier-Legendre expansion of the scattering phase function (SPF) and provides the updated framework for computing skylight and sunlight inscattering.

- Moved original Section 3.1.3, describing how we model solar irradiance, into Appendix A (Section A.2).

- Moved original Section 3.1.4, describing the truncation of the Fourier-Legendre-expanded SPF, into Appendix A (Section A.1) – see also our response to the Referee's last minor comment.

- Shortened original Section 3.2.1 and merged it with original Section 3.2 – see also our response to the Referee's first minor comment.

We have also revised text in Section 3 to identify that the presented theory represents the current standard approach to analyzing sky-LOSA data; specifically, "[t]his section describes the MC method used in the present [general uncertainty analysis] including novel updates to the MC approach that are necessary to make this present work tractable and significantly accelerate future sky-LOSA analyses."

***Length (continued).*** *Similarly, too many acronyms are used in these sections and are not used frequently enough to be necessary (including ET, SPF, CM-LHS, ...) and not all symbols are defined next to their equations (e.g. L(b) in Equation 9 and ak next to Equation 18).*

We have deleted unnecessary acronyms in the manuscript and clarified symbols/nomenclature.

***Monte Carlo clarification.*** *A Monte Carlo calculation randomly samples prior distribution(s) and repeats a calculation in order to obtain a posterior distribution of results. The key question here is what priors were assumed, and how accurate are they? The manuscript glosses over this point and takes the MC output as correct without any top-down validation. Please revise Section 3 and Table 2 to emphasize the prior distributions used. The authors have already done this in their earlier work (Johnson et al., 2013, Table 2) by tabulating "Distributions used in MC". I believe the authors did intend to include this information but I do not find it clear enough.*

Prior distributions for the eight fundamental soot properties required by the sky-LOSA method were derived from a thorough literature review of soot generated from flare-like flames. We originally referenced the source for these distributions (Johnson et al., 2013) but have now included them in the updated Table 2 for completeness. Further to the comments of Referee #3, we also now list the properties at the first mention of vector ***b*** in Section 2.

***Monte Carlo clarification (continued).*** *In Table 2 of this work, the last column "MC Implementation" is specified as "MC-randomized" multiple times – this is a meaningless statement. Of course the MC calculation performs random sampling.*

We have removed the term "MC- (Monte Carlo)-randomized" and instead refer to variables perturbed within MC analyses as *random* variables.

***Validation of assumptions.*** *The manuscript assumes throughout that a perfect skyLOSA measurement gives a perfect result. This has not been justified in the manuscript nor in earlier skyLOSA work, to my knowledge. These calculations are not constrained by any direct measurements. The skyLOSA approach is comparable to a satellite retrieval algorithm and requires direct validation. Until directly validated, this limitation must always be repeated.*

In the case of sky-LOSA, there is no comparable quantitative reference standard for time-resolved soot/BC emissions from open flames like gas flares; the current measurement standard is an assessment of plume opacity by a human observer (U.S. EPA, 1974). However, significant validation work of the sky-LOSA approach and underlying assumptions has indeed been performed using a range of alternate approaches, as necessary.

The novel aspect of sky-LOSA is the quantification of soot/BC mass column density along a line-of-sight traversing a plume using radiometric observations coupled with radiative transfer considerations. As the enabling component of sky-LOSA, this theory (presented in Section 2 of the original manuscript) has been the focus of a range of validation efforts. As referenced in the first line of Section 2, the first-generation of the sky-LOSA approach was validated by Johnson et al. (2010) against a commercial laser-induced incandescence instrument and the proven lab-based diffuse-LOSA measurement (Thomson et al., 2008) on which the technique is ultimately based. Proof-of-concept field measurements were subsequently performed in Uzbekistan (Johnson et al., 2011). Johnson et al., (2013) then extended the earliest theory to consider the effect of inscattering of skylight and sunlight that bias the perceived opacity of the plume and, hence, emission rate. In this same work, they derived the prior distributions of soot/BC properties used in the present sky-LOSA method to bound the

influence of uncertain soot/BC properties on computed emissions. This theory was used in field measurements in Mexico and Ecuador (Conrad and Johnson, 2017; Johnson et al., 2013).

At this point, sky-LOSA had been validated in its ability to quantify soot/BC mass loading and the theory developed to robustly consider uncertain soot/BC properties and inscattering effects. Validation efforts then turned to two final radiative transfer considerations that could influence sky-LOSA's ability to accurately quantify soot/BC loading. Conrad et al. (2020a) leveraged large-eddy simulations of flares to prove that the effect of refractive index gradient-driven beam steering in flare plumes was negligible in the visible spectrum where sky-LOSA data are acquired. Then, Conrad et al. (2020b) amended the sky-LOSA theory to consider the minor/second-order effect of multiple scattering on the quantification of inscattered light in sky-LOSA analyses. From the perspective of first principles, these analyses completed our efforts to validate the sky-LOSA technique.

To direct the interested reader to these substantial validation efforts, we have revised the first line of Section 2 to reference validation-related publications.

***Validation of assumptions (continued).*** *The concentrations reported by the current approach are a type of "equivalent black carbon" defined by the authors' assumptions.*

As discussed in the seminal works of Andreae and Gelencsér (2006) and Petzold et al. (2013), light absorption measurements of "black carbon" mass are inherently defined by the assumed optical properties of the absorbing particulate. These manuscripts both suggest use of the term "equivalent black carbon" for absorption-based techniques although this term has not been broadly adopted within the literature. Instead, it is generally implied that BC measured using such optical techniques is inherently "equivalent BC".

Thus, we absolutely agree with the Referee and have revised the text in Section 2 to specifically note that sky-LOSA-reported mass emissions are dependent upon the prior soot/BC property distributions:

> *"..., it is worth noting that these prior distributions of soot/BC properties inherently link light absorption measurements and computed mass column density/emissions using sky-LOSA. Thus, in keeping with Andreae and Gelencsér (2006) and Petzold et al., (2013), sky-LOSA-inferred soot/BC mass might be called 'equivalent BC' as is recommended for all light absorption-based diagnostics."*

***Validation of assumptions (continued).*** *My main concern here is with respect to the aerosol optical properties, which have not been discussed at all. Instead, Johnson et al. 2013 is cited. The authors have assumed that the flame emits only soot. What about organics, which may condense when the plume cools? The photograph in Johnson et al. 2013 clearly suggests that the plume may have cooled before measurement. What about inorganics such as sulfates? How pure are the fuels burnt in these flares? Any impurities are likely to influence the aerosol optical properties. Please add calculations where black carbon is assumed to be mixed with organics or other impurities, using reasonable and literature-based assumptions, and show how the conclusions of this work change in response.*

The Referee's main concern is that the presence of non-BC material in flare plumes could bias sky-LOSA-computed soot/BC emissions by their influence on aerosol optical properties. While this is somewhat at odds with the notion of "equivalent BC" (see the Referee's previous comment) – i.e., inferred mass is *operationally defined* by the assumed optical properties – numerous observations in the literature show that non-BC material does not contribute to optical observations of fresh flare plumes in the visible spectrum.

In theory, both internal and external mixtures of soot/BC with other material could bias sky-LOSA-calculated emissions by changing the optical properties of particulate in the flare plume:

1.  If emitted soot/BC were internally mixed/coated with a scattering material (via condensation of co-emitted organic species, for example), then emitted BC could have enhanced absorption (e.g., Bond et al., 2006).

2.  Alternatively, if emitted soot/BC were externally mixed with significant co-emitted primary (and secondary) aerosols, the effective absorption/scattering by the plume could be different than that of soot/BC alone.

For the specific case of gas flares, it has been observed both in the laboratory and the field that optically active material in the visible spectrum in flare plumes is solely soot/BC. First, in their field measurements in the Bakken region, Schwarz et

al., (2015) showed that "flaring BC was not associated with optically significant internally mixed non-BC material or with significant emissions of non-BC-containing primary aerosol". Similarly, Weyant et al., (2016) concluded in their field measurements that the presence of non-BC aerosols in a flare plume is "not statistically different from zero". Importantly, sky-LOSA measurements are performed in the very near-field of the flame – e.g., the noted Fig. 2 in Johnson et al. (2013) shows sky-LOSA measurement of emissions over a control surface that is within meters of the flame tip. While the plume has indeed cooled at this location (likely within 200 K of ambient temperature (Poudenx, 2000)), the soot/BC particulate is very fresh relative to typical atmospheric soot/BC that may be internally mixed with other materials. In fact, the aforementioned field studies were performed well downstream of the flame (at the basin level in the case of Schwarz et al., (2015) and hundreds of meters downstream in the case of Weyant et al., (2016)), where particles would have been subject to much more atmospheric aging and thus much more likely to be mixed with non-BC material.

These field observations are supported by laboratory studies of freshly emitted soot/BC from large flare-like flames burning fuels representative of global oil and gas flaring. Specifically, electron micrographs from Kazemimanesh et al. (2019) identify that co-emitted organics are not present as a coating on flare-emitted soot/BC. This is likely because emitted organics are largely unburned fuel (Johnson et al., 2001) which are dominated by highly volatile, low molecular weight hydrocarbons (e.g., Conrad and Johnson, 2019).

Thus, these literature data conclusively show that fresh flare particulate as measured by sky-LOSA is soot/BC, which supports our use of the literature-derived flare soot/BC properties from Johnson et al. (2013).

**Validation of assumptions (continued).** *A lesser concern is the assumption (Section 3.1.3) of an ideal clean atmosphere. What about background aerosol? Surely the air around an oil field is not perfectly clean.*

The sky-LOSA method does not inherently assume an ideal clean atmosphere. The original Section 3.1.3 (now Appendix Section A.2) discussed how we model ground-level solar irradiance in the context of the sky-LOSA Monte Carlo method. We believe that the Referee is referring to the "ideal extinction for a clean atmosphere at a given relative air mass" ($\sigma^{e*}(m)$) that is used to compute the ground-level solar irradiance from the extra-terrestrial solar irradiance. Importantly, this ideal extinction is multiplied by the sky-dependent turbidity factor ($T(a)$) to represent extinction through a non-ideal (i.e., polluted) atmosphere. That is, an ideal clean atmosphere is **not** assumed, but is scaled to represent a range of realistic atmospheric conditions. To clarify this in the revised manuscript, we have included the new text (in bold):

> "...$\sigma^{e*}$ is the ideal extinction for a clean atmosphere at a given relative air mass, and $T(a)$ is the model-dependent turbidity factor **that is used to consider realistic atmospheres and describes** the number of clean atmospheres required to represent **attenuation through the non-ideal, polluted atmosphere**."

**Minor Comments:**

*The justification of a quantile-based coefficient of variation in Section 3.2.1 can be shortened.*

We have shortened this section as requested and merged it into the preceding section, 3.2.

*The word compiled in Section 3.1.5 should probably be changed to grouped. And I am not sure I understand what concept the authors are trying to convey here. Was the grouping done based on skyLOSA results?*

We have removed the word "compiled" and amended our original notation to say "sorted into sky 'categories'".

In our original text, we present the concept (emphasis added) that "...there is some additional uncertainty in sky-LOSA-computed soot mass column density through use of a *single* CIE sky model in the MC method" due to the inherent error in these simple models. We also present how the sky categories were defined (emphasis added): "...to permit capture of CIE sky model error in the [general uncertainty analysis], like skies were [sorted] into sky '[categories]' *that have similar properties but differing … directional variability*." Broad descriptions and characteristics of each sky category are located in the final paragraph of Section 3.1.2 and Table 1 of the revised manuscript.

*I found the discussion of the total order L(b) in Section 3.1.4 unclear. Is this discussion significant, considering the uncertainties in the assumption of a black-carbon-only aerosol and aerosol-free sky?*

As noted in the responses above, we do not assume an aerosol-free sky.

This brief section describes our approach to the truncation of the Fourier-Legendre expansion of the scattering phase function; specifically, the order ($L(\boldsymbol{b})$) at which the expansion was truncated while both ensuring accuracy in sky-LOSA-computed soot/BC mass column density and enabling rapid calculation of sky- and sunlight inscattering. To help reduce the length of the main manuscript (see the Referee's first general comment), we have moved this text to the new Appendix A (Section A.1).

**References**

Andreae, M. O. and Gelencsér, A.: Black carbon or brown carbon? The nature of light-absorbing carbonaceous aerosols, Atmos. Chem. Phys., 6(10), 3131–3148, doi:10.5194/acp-6-3131-2006, 2006.

Bond, T. C., Habib, G. and Bergstrom, R. W.: Limitations in the enhancement of visible light absorption due to mixing state, J. Geophys. Res., 111(D20), D20211, doi:10.1029/2006JD007315, 2006.

Conrad, B. M. and Johnson, M. R.: Field measurements of black carbon yields from gas flaring, Environ. Sci. Technol., 51(3), 1893–1900, doi:10.1021/acs.est.6b03690, 2017.

Conrad, B. M. and Johnson, M. R.: Mass absorption cross-section of flare-generated black carbon: Variability, predictive model, and implications, Carbon N. Y., 149, 760–771, doi:10.1016/j.carbon.2019.04.086, 2019.

Conrad, B. M., Thornock, J. N. and Johnson, M. R.: Beam steering effects on remote optical measurements of pollutant emissions in heated plumes and flares, J. Quant. Spectrosc. Radiat. Transf., 254, doi:10.1016/j.jqsrt.2020.107191, 2020a.

Conrad, B. M., Thornock, J. N. and Johnson, M. R.: The effect of multiple scattering on optical measurement of soot emissions in atmospheric plumes, J. Quant. Spectrosc. Radiat. Transf., 254, 107220, doi:10.1016/j.jqsrt.2020.107220, 2020b.

Johnson, M. R., Wilson, D. J. and Kostiuk, L. W.: A fuel stripping mechanism for wake-stabilized jet diffusion flames in crossflow, Combust. Sci. Technol., 169(1), 155–174, doi:10.1080/00102200108907844, 2001.

Johnson, M. R., Devillers, R. W., Yang, C. and Thomson, K. A.: Sky-Scattered solar radiation based plume transmissivity measurement to quantify soot emissions from flares, Environ. Sci. Technol., 44(21), 8196–8202, doi:10.1021/es1024838, 2010.

Johnson, M. R., Devillers, R. W. and Thomson, K. A.: Quantitative field measurement of soot emission from a large gas flare using sky-LOSA, Environ. Sci. Technol., 45(1), 345–350, doi:10.1021/es102230y, 2011.

Johnson, M. R., Devillers, R. W. and Thomson, K. A.: A generalized sky-LOSA method to quantify soot/black carbon emission rates in atmospheric plumes of gas flares, Aerosol Sci. Technol., 47(9), 1017–1029, doi:10.1080/02786826.2013.809401, 2013.

Kazemimanesh, M., Dastanpour, R., Baldelli, A., Moallemi, A., Thomson, K. A., Jefferson, M. A., Johnson, M. R., Rogak, S. N. and Olfert, J. S.: Size, effective density, morphology, and nano-structure of soot particles generated from buoyant turbulent diffusion flames, J. Aerosol Sci., 132, 22–31, doi:10.1016/j.jaerosci.2019.03.005, 2019.

Petzold, A., Ogren, J. A., Fiebig, M., Laj, P., Li, S.-M., Baltensperger, U., Holzer-Popp, T., Kinne, S., Pappalardo, G., Sugimoto, N., Wehrli, C., Wiedensohler, A. and Zhang, X.-Y.: Recommendations for reporting "black carbon" measurements, Atmos. Chem. Phys., 13(16), 8365–8379, doi:10.5194/acp-13-8365-2013, 2013.

Poudenx, P.: Plume sampling of a flare in crosswind: structure and combustion efficiency, M.Sc. Thesis, University of Alberta, Edmonton, AB, Canada, Edmonton. [online] Available from: http://www.collectionscanada.gc.ca/obj/s4/f2/dsk1/tape4/PQDD_0011/MQ59867.pdf, 2000.

Schwarz, J. P., Holloway, J. S., Katich, J. M., McKeen, S., Kort, E. A., Smith, M. L., Ryerson, T. B., Sweeney, C. and Peischl, J.: Black carbon emissions from the Bakken oil and gas development region, Environ. Sci. Technol. Lett., 2(10), 281–285, doi:10.1021/acs.estlett.5b00225, 2015.

Thomson, K. A., Johnson, M. R., Snelling, D. R. and Smallwood, G. J.: Diffuse-light two-dimensional line-of-sight attenuation for soot concentration measurements, Appl. Opt., 47(5), 694–703, doi:10.1364/AO.47.000694, 2008.

U.S. EPA: Visual Determination of the Opacity of Emissions from Stationary Sources, Code of Federal Regulations, Title 40, Part 60, Appendix A-4, Method 9, United States of America., 1974.

Weyant, C. L., Shepson, P. B., Subramanian, R., Cambaliza, M. O. L. L., Heimburger, A., Mccabe, D., Baum, E., Stirm, B. H. and Bond, T. C.: Black carbon emissions from associated natural gas flaring, Environ. Sci. Technol., 50(4), 2075–2081, doi:10.1021/acs.est.5b04712, 2016.

---

## Author Comment (AC2) · 18 Dec 2020

**Manuscript ID:** amt-2020-255
**Title:** An uncertainty-based protocol for the setup and measurement of soot/black carbon emissions from gas flares using sky-LOSA
**Authors:** Conrad, Bradley M.; Johnson, Matthew R.

**Point-by-point Response to Comments by Referee #3**

*This manuscript presented an uncertainty-based guide/instruction for potential users of the sky-LOSA technique to measure soot/black carbon emissions from flares in the oil and gas industry. Although the method itself and the details of various models involved in the data analysis have been published by the authors, the paper is useful to help potential researchers/engineers to better understand and apply this promising technique. The paper is well written as deserves publication. I have few questions/comments for the authors to consider.*

We thank the Referee for their positive comments and recognition of the value of the manuscript and software tool to help others to "better understand and apply this promising technique".

*1. In line 29 on page 4, please explicitly list the eight soot properties.*

We have appended a list of these eight soot properties to the noted paragraph. Further to the comments of Referee #1, we have also included the properties and their distributions in Table 2 of the revised manuscript.

*2. In the derivation of Eq. (4), do you need to assume that the mass-normalized extinction cross section of soot is constant everywhere in the flare? Otherwise, it cannot be taken out of the second integral on the right-hand side of Eq. (4).*

To remove the mass-normalized extinction cross-section from this integral we must assume that it does not vary in space. This assumption is supported by spatially resolved measurements of soot properties within the literature. For example, Köylü and Faeth's (1992) measurements of soot morphology in the overfire region of large-scale buoyancy-driven turbulent diffusion flames have shown that "the structure of soot … is relatively independent of … position in the overfire region" – and stated more conclusively in a later publication (1994): "…soot structure is independent of position in the overfire region…". We have clarified this assumption in Section 2 of the revised manuscript.

Although we inherently assume spatiotemporal uniformity of the soot properties, it is important to note that these soot properties are independently perturbed within sky-LOSA's Monte Carlo analysis to bound the influence of their uncertainty on sky-LOSA-computed emissions.

*3. In this work, the authors dealt with the in-scattering term by expanding the SPF in Fourier-Legendre series. As the authors admitted, this expansion requires a large number of terms to have an accurate representation of the SPF, especially when it is highly forward peaked. There are other methods to more efficiently deal with highly-forward peaked SPF, such as the Henyey-Greenstein and transport approximations, have the authors considered such approaches in the calculations of the in-scattering terms?*

Although there are a few different options for modelling scattering phase functions (SPFs), the Fourier-Legendre expansion has the advantage of being numerically exact, even if more computationally demanding. Moreover, while not as efficient as the HG phase function, calculation of the Legendre coefficients was quite rapid when following the approach of Schuster (2004) such that their computation was negligible compared to the execution of the Monte Carlo method of the present general uncertainty analysis.

Early in this research we did consider the Henyey-Greenstein (HG) phase function as suggested by the reviewer, but ultimately decided that it was not suitable for the sky-LOSA technique. To elaborate, the HG phase function is a simple one-parameter model that approximates the soot/Legendre coefficients via $\Phi_l = g_{HG}^l(2l + 1)$ (e.g., Boucher, 1998), where $g_{HG}$ is the "anisotropy factor" of the phase function that is usually estimated via least-squares fitting to the RDG-computed SPF (e.g., Daun et al., 2008). While the HG phase function renders calculation of the soot coefficients trivial, our experience has shown that the single-parameter HG phase function may not always capture the highly asymmetric SPFs that are typical of soot, especially larger aggregate populations. The below figure plots the angle-resolved error of the HG phase function for a range of soot properties obtained using the prior distributions in the manuscript – here, the colour of each plot represents the asymmetry parameter of the soot population. The figure shows that errors can exceed 50% but, more

importantly and perhaps unsurprisingly, these errors tend to be highest for large aggregates where the SPF is more asymmetric. This implies that the HG phase function is a poorer alternative in the context of sky-LOSA, where measurement uncertainty is largely controlled by the forward scattering of sky- and sunlight into the camera (as discussed in the text surrounding Fig. 4 in the original manuscript).

[Figure]

*4. In this paper, the authors did not provide the detailed soot properties, but reference to a previous study. It appears that the authors assume that the soot properties are uniform over the entire flare under consideration and also remain the same from one flare to another. Is this correct? If yes, the authors need to justify this assumption. The soot properties in a flare with soot emission reduction measure (such as partial premixing) may not be the same as those without any such measures.*

While we defensibly assume spatiotemporal uniformity of soot properties *within a given draw* of the Monte Carlo (MC) analysis (see the Referee's second comment), it is important to note that these soot properties are perturbed within the MC method over notably broad prior distributions. These prior distributions were derived from an exhaustive literature review of soot property data (Johnson et al., 2013), including studies of soot generated by flames of varying scales and configurations (premixed/non-premixed) burning fuels that extend well beyond typical flare gas compositions. Despite the likely variability of soot properties from flare to flare, the breadth of these priors suggests that sky-LOSA confidence intervals are likely to bound the ground truth emission rate; for this reason, the same prior distributions of soot properties are used in all sky-LOSA analyses.

*5. The choice of sky model group seems not straightforward and has a strong influence on the measurement. I wonder if the authors can provide more useful 'tips' to make such a choice for new users of sky-LOSA.*

Sky categories/groups are selected using criteria laid out in the rightmost column of Table 1. These criteria are based on simple observations by the sky-LOSA user such as whether the sun is obstructed or unobstructed and whether the sky is overcast, partly cloudy, or clear. Sky-LOSA-calculated soot emissions are indeed sensitive to the selected CIE sky model; however, *within* each of the defined sky categories/groups (e.g., highly turbid overcast skies (category/group "A")), different sky models generally have similar effects on computed emissions and uncertainties. In fact, it is this weak variation in the effect of the models within a sky category/group that allows us to define the sky categories/groups and ultimately treat the sky model as a random parameter within the MC analysis with minor impact on measurement uncertainty.

To clarify how a user might select the most appropriate sky category/group, we have added text to Section 3.1.2 of the revised manuscript noting the simple selection criteria.

**References**

Boucher, O.: On Aerosol Direct Shortwave Forcing and the Henyey–Greenstein Phase Function, J. Atmos. Sci., 55(1), 128–134, doi:10.1175/1520-0469(1998)055<0128:OADSFA>2.0.CO;2, 1998.

Daun, K. J., Thomson, K. A. and Liu, F.: Simulation of Laser-Induced Incandescence Measurements in an Anisotropically

Scattering Aerosol Through Backward Monte Carlo, J. Heat Transfer, 130(11), 1–10, doi:10.1115/1.2955468, 2008.

Johnson, M. R., Devillers, R. W. and Thomson, K. A.: A generalized sky-LOSA method to quantify soot/black carbon emission rates in atmospheric plumes of gas flares, Aerosol Sci. Technol., 47(9), 1017–1029, doi:10.1080/02786826.2013.809401, 2013.

Köylü, Ü. Ö. and Faeth, G. M.: Structure of overfire soot in buoyant turbulent diffusion flames at long residence times, Combust. Flame, 89(2), 140–156, doi:10.1016/0010-2180(92)90024-J, 1992.

Köylü, Ü. Ö. and Faeth, G. M.: Optical properties of overfire soot in buoyant turbulent diffusion flames at long residence times, J. Heat Transfer, 116(1), 152–159, doi:10.1115/1.2910849, 1994.

Schuster, G. L.: Inferring the Specific Absorption and Concentration of Black Carbon From Aeronet Aerosol Retrievals, Pennsylvania State University., 2004.

---

## Author Response (AR2)

**Manuscript ID:** amt-2020-255
**Title:** An uncertainty-based protocol for the setup and measurement of soot/black carbon emissions from gas flares using sky-LOSA
**Authors:** Conrad, Bradley M.; Johnson, Matthew R.

**Point-by-point Response to Comments by Referee**

*The authors Conrad and Johnson have addressed most of my feedback. I only have two comments remaining.*

*Length: the authors responded to my request to shorten the manuscript by stating various justifications. These should be added to the text with one or two introductory sentences. The same applies to the authors' response to my comment on the presence of non-BC impurities. Please add a brief summary to the main text.*

1) We have revised the introductory paragraph to Section 3 ("General uncertainty analysis methodology") to highlight our justification for the comprehensive methodology section, and specifically that "[t]his new methodology is a significant improvement to the sky-LOSA algorithm that enables accelerated MC-computation of soot column density and, hence, emission rates from sky-LOSA image data".

2) We have included additional text to the paragraph following Equation (3) to highlight that absorption-enhancing non-BC material is not likely to be observed in flare plumes, especially in the near field, according to both laboratory (Kazemimanesh et al., 2019) and field data (Schwarz et al., 2015; Weyant et al., 2016), which justifies the literature-derived probability distributions used in sky-LOSA (Johnson et al., 2013).

*Validation of assumptions: the authors' response to my comment ("The manuscript assumes throughout that a perfect skyLOSA measurement gives a perfect result") was "sky-LOSA has been validated [...] from the perspective of first principles." The authors full response included a restatement of the bottom-up work done so far. This work is extensive and thorough, especially in the context of atmospheric measurement techniques, but does not address my comment. The purpose of my comment is to highlight that bottom-up work requires top-down validation (to the extent that such is possible).*

*For example, Thomson et al. (2004, https://doi.org/10.1016/j.combustflame.2004.11.012) state that "The uncertainty of the LOSA soot volume fraction measurements is estimated to be 20 to 30% (95% confidence interval). The uncertainty is dominated by the uncertainties in the magnitude of and the contribution of scatter to light attenuation measurements." So the authors may add a statement such as "The LOSA technique is estimated to be 20 to 30% accurate. We estimate that the calculation of the inscattering of solar radiation adds an additional X\% to this uncertainty using our MC technique, resulting in an overall uncertainty of Y%." Please add such a statement to the manuscript.*

Overall measurement uncertainties vary with field conditions and equipment setup, which is indeed the motivation behind this paper. For most real-world field conditions, overall sky-LOSA uncertainties are in the range of −26/+36% (e.g., Johnson et al., 2013) – we have added text to specifically note this in the penultimate paragraph of Section 2 where the sky-LOSA method is described. Importantly, as detailed in the manuscript, sky-LOSA uncertainties are rigorously calculated on a case-by-case basis via a Monte Carlo method, where fundamental soot properties and field conditions (sky and solar radiation) are treated as random variables. This contrasts with the cited earlier work of Thomson et al., (2005), where uncertainties in their laboratory-based collimated-LOSA technique were estimated by summing selected uncertainties in quadrature (Thomson, 2004): $E(m_\lambda)$, the scatter-to-absorption ratio (related to the single-scatter albedo, $\omega(\boldsymbol{b})$), and diagnostic-specific measurement uncertainties that are unrelated to sky-LOSA.

We agree with the Referee that it is important to validate bottom-up measurements with a top-down alternative to the extent that such is possible. The challenge – as noted by the Referee – is that top-down validation can only be performed if a satisfactory alternative measurement technique exists. Echoing our first response to the Referee, this is unfortunately not the case for sky-LOSA since the current measurement "standard" for flare soot/BC emissions is a visible assessment of plume opacity by a human observer (U.S. EPA, 1974). There are aircraft-based (top-down) techniques that use atmospheric measurements of BC and combustion product concentrations to infer flare BC emissions based on an assumed flare gas composition (Gvakharia et al., 2017; Schwarz et al., 2015; Weyant et al., 2016). However, because BC emission rates are very strongly dependent on fuel composition (e.g., McEwen and Johnson, 2012), and due to the stochastic nature of flare BC emissions (Conrad and Johnson, 2017), it is not obvious how these approaches can resolve robust emission rate statistics sufficient to serve as an additional validation for sky-LOSA.

**References**

Conrad, B. M. and Johnson, M. R.: Field measurements of black carbon yields from gas flaring, Environ. Sci. Technol., 51(3), 1893–1900, doi:10.1021/acs.est.6b03690, 2017.

Gvakharia, A., Kort, E. A., Brandt, A. R., Peischl, J., Ryerson, T. B., Schwarz, J. P., Smith, M. L. and Sweeney, C.: Methane, black carbon, and ethane emissions from natural gas flares in the Bakken Shale, North Dakota, Environ. Sci. Technol., 51(9), 5317–5325, doi:10.1021/acs.est.6b05183, 2017.

Johnson, M. R., Devillers, R. W. and Thomson, K. A.: A generalized sky-LOSA method to quantify soot/black carbon emission rates in atmospheric plumes of gas flares, Aerosol Sci. Technol., 47(9), 1017–1029, doi:10.1080/02786826.2013.809401, 2013.

Kazemimanesh, M., Dastanpour, R., Baldelli, A., Moallemi, A., Thomson, K. A., Jefferson, M. A., Johnson, M. R., Rogak, S. N. and Olfert, J. S.: Size, effective density, morphology, and nano-structure of soot particles generated from buoyant turbulent diffusion flames, J. Aerosol Sci., 132, 22–31, doi:10.1016/j.jaerosci.2019.03.005, 2019.

McEwen, J. D. N. and Johnson, M. R.: Black Carbon Particulate Matter Emission Factors for Buoyancy Driven Associated Gas Flares, J. Air Waste Manage. Assoc., 62(3), 307–321, doi:10.1080/10473289.2011.650040, 2012.

Schwarz, J. P., Holloway, J. S., Katich, J. M., McKeen, S., Kort, E. A., Smith, M. L., Ryerson, T. B., Sweeney, C. and Peischl, J.: Black carbon emissions from the Bakken oil and gas development region, Environ. Sci. Technol. Lett., 2(10), 281–285, doi:10.1021/acs.estlett.5b00225, 2015.

Thomson, K. A.: Soot Formation in Annular Non-premixed Laminar Flames of Methane-Air at Pressures of 0.1 to 4.0 MPa, University of Waterloo., 2004.

Thomson, K. A., Gülder, Ö. L., Weckman, E., Fraser, R., Smallwood, G. J. and Snelling, D. R.: Soot concentration and temperature measurements in co-annular, nonpremixed CH4/air laminar flames at pressures up to 4 MPa, Combust. Flame, 140(3), 222–232, doi:10.1016/j.combustflame.2004.11.012, 2005.

U.S. EPA: Visual Determination of the Opacity of Emissions from Stationary Sources, Code of Federal Regulations, Title 40, Part 60, Appendix A-4, Method 9, United States of America., 1974.

Weyant, C. L., Shepson, P. B., Subramanian, R., Cambaliza, M. O. L. L., Heimburger, A., Mccabe, D., Baum, E., Stirm, B. H. and Bond, T. C.: Black carbon emissions from associated natural gas flaring, Environ. Sci. Technol., 50(4), 2075–2081, doi:10.1021/acs.est.5b04712, 2016.